# KL Guided Domain Adaptation

**A. Tuan Nguyen**[*][†]          **Toan Tran** [‡]          **Yarin Gal** [†]

**Philip H. S. Torr** [†]          **Atılım Güneş Baydin** [†]

## Abstract

Domain adaptation is an important problem and often needed for real-world applications. In this problem, instead of i.i.d. training and testing datapoints, we assume that the source (training) data and the target (testing) data have different distributions. With that setting, the empirical risk minimization training procedure often does not perform well, since it does not account for the change in the distribution. A common approach in the domain adaptation literature is to learn a representation of the input that has the same (marginal) distribution over the source and the target domain. However, these approaches often require additional networks and/or optimizing an adversarial (minimax) objective, which can be very expensive or unstable in practice. To improve upon these marginal alignment techniques, in this paper, we first derive a generalization bound for the target loss based on the training loss and the reverse Kullback-Leibler (KL) divergence between the source and the target representation distributions. Based on this bound, we derive an algorithm that minimizes the KL term to obtain a better generalization to the target domain. We show that with a probabilistic representation network, the KL term can be estimated efficiently via minibatch samples without any additional network or a minimax objective. This leads to a theoretically sound alignment method which is also very efficient and stable in practice. Experimental results also suggest that our method outperforms other representation-alignment approaches.

## 1 Introduction

With advances in neural network architectures (He et al., 2016; Vaswani et al., 2017), machine learning algorithms have achieved state-of-the-art performance in many tasks such as object classification, object detection and natural language processing. However, machine learning models have been focusing mostly on the case of independent and identically distributed (i.i.d.) datapoints; and such an assumption often does not hold in practice. When the i.i.d. assumption is violated and the target domain has a different distribution compared to the source domain, a typical learner trained on the source data via empirical risk minimization would not perform well at test time, since it does not account for the distribution shift. To tackle this problem, many methods have been proposed for domain adaptation (Zhao et al., 2019; Zhang et al., 2019; Combes et al., 2020; Tanwani, 2020) and domain generalization (Khosla et al., 2012; Muandet et al., 2013; Ghifary et al., 2015), the goal of which is to train a machine learning algorithm that can generalize well to the target domain.

A common approach to tackle these problems is to learn a representation such that its distribution does not change across domains. There are two types of distribution alignment: marginal alignment (aligning the marginal distribution of the representation) and conditional alignment (aligning the conditional distribution of the label given the representation) (Nguyen et al., 2021; Tanwani, 2020). For domain adaptation and domain generalization problems with multiple source domains, we can use the data and labels to align both the marginal and the conditional distributions across the source domains, aiming to generalize to the target domain. However, in a single-source domain adaptation problem, with only unlabeled data from the target domain, it is often only possible to align the marginal distribution of the representation. This marginal alignment should help the classifier avoid out-of-distribution data at test time.

---

[*]Corresponding author: A. Tuan Nguyen, `tuan@robots.ox.ac.uk`

[†]University of Oxford, Oxford, United Kingdom

[‡]VinAI Research, Hanoi, Vietnam

This paper focuses on such a single-source domain adaption problem, which is also one of the most common settings in practice. Current marginal alignment techniques usually require additional computation (e.g., of an additional network) (Ganin et al., 2016; Li et al., 2018) and/or a minimax objective (Ganin et al., 2016; Shen et al., 2018), leading to an expensive and/or unstable training procedure (Goodfellow, 2016; Kodali et al., 2017). For example, DANN (Ganin et al., 2016) employs an adversarial training procedure, with a domain discriminator that classifies the domain of the representation, and maximizes the adversarial loss of the discriminator. When the discriminator is completely fooled, the marginal distribution of the representation is aligned across domains. MMD (Gretton et al., 2012; Li et al., 2018) utilizes maximum mean discrepancy to align the representation distribution. This does not use a minimax objective, thus leading to a more stable training; however, it does require additional computation of several Gaussian kernels. While more sophisticated (non-marginal-alignment) methods have been proposed recently to achieve better results in the domain adaptation problem, we argue that studying the family of plain marginal-alignment techniques is still an important task, since they are the backbones that most other domain adaptation (and domain generalization) methods are built upon.

To address the above issues of existing marginal-alignment techniques, we first derive a generalization bound on the loss of the target domain using the training loss and a reverse Kullback–Leibler (KL) divergence between the source and target distributions. There are existing bounds of the target loss in the literature (Ben-David et al., 2010), however, these analyses focus mostly on the case of binary classification and the bounds use a total variation distance or a $\mathcal{H}$-divergence between the distributions, which are not easy to estimate in practice (for example, Ajakan et al. (2014) require an adversarial network to estimate the $\mathcal{H}$-divergence). In this paper, we show that with a probabilistic representation network, we can estimate the KL divergence easily using samples, leading to an alignment method that requires virtually no additional computation nor a minimax objective. Therefore, our training procedure is simple and stable in practice. Moreover, the reverse KL has the zero-forcing effect (Minka et al., 2005), which is very effective to alleviate the out-of-distribution problem in practice. This can be explained as follows: the out-of-distribution problem arises when the classifier faces a new representation at test time that is in a (near) zero mass region of the source representation distribution (and thus it never faced before). The reverse KL tends to force the target representation distribution to have (near) zero mass wherever the source distribution has (near) zero mass (this is the zero-forcing property), which helps the classifier avoid out-of-distribution data. The reverse KL also has the mode-seeking effect (Minka et al., 2005) which allows for a more flexible alignment of the representation (to one or some of the modes of the source domain). For example, consider the classification problem of buildings (houses, hotels, etc.) where source images are collected from urban and remote areas of a country (two modes); while the target images are collected from urban areas but from a different country. Ideally, we want to match the representation distribution of the target domain to that of the first mode of the source domain since they are both from urban areas. The reverse KL allows this flexible alignment (as it results in a relatively small value of the reverse KL) due to its mode-seeking property. Meanwhile, other distance metrics/divergences aim to match the whole source and target representation distribution, which might collapse the two modes of the source domains.

Our contributions in this work are:

- We construct a generalization bound of the test loss in the domain adaptation problem using the reverse KL divergence.
- We propose to reduce the generalization bound by minimizing the above KL term. Furthermore, we show that with a probabilistic representation, the KL term can be estimated easily using minibatches, without any additional computation or a minimax objective as opposed to most existing works.
- We conduct extensive experiments and show that our method significantly outperforms relevant baselines, namely ERM (Bousquet et al., 2003), DANN (Ganin et al., 2016), MMD (Gretton et al., 2012; Li et al., 2018), CORAL (Sun and Saenko, 2016) and WD (Shen et al., 2018). We empirically show that the reverse KL divergence is very effective for representation alignment since it is very stable and efficient to compute in practice.

## 2 RELATED WORK

**Generalization bound for the distribution shift problem**    There exist works studying bounds for the distribution shift problem in the literature (Ben-David et al., 2010; Mansour et al., 2009). How-

ever, their analyses of the classification problem are limited to the case of binary labels. Moreover, these bounds are only applicable or practical for deterministic labeling functions, which is not the case for most datasets in practice. Therefore, their analyses cannot be generalized to the general case of supervised learning. The differences between our bound and theirs are as follows. First of all, our bound works for the general case of supervised learning: it works for both the classification (including multiclass classification) and regression problems, it makes no assumptions about the labeling mechanism (can be probabilistic or deterministic), and it works for virtually all predictive distributions commonly used in practice. Secondly, our bound uses a different divergence, namely the KL divergence, which is easier to estimate in practice compared to total variation or $\mathcal{H}$-divergence. We provide a brief review of the above bounds and discuss their differences to ours in more detail in the appendix. Recently, Acuna et al. (2021) revise the previous domain adaptation bounds and generalize them to a multi-class classification setting, as well as to the class of $f$-divergence (including KL divergence). However, their setting is still very restricted: the loss function needs to satisfy the triangle inequality (which does not hold for many loss functions in practice), and they need to know the true labeling function (optimal Bayes classifier) for a probabilistic labeling mechanism (which is often not available). We also provide further analysis, that under some reasonable assumptions, the conditional misalignment in the representation space is bounded by the conditional misalignment in the input space, which allows for a sound marginal alignment method. This can be seen as an improvement over prior works. Some specific cases of distribution shift have also been studied. For example, Cortes et al. (2010) and Johansson et al. (2019) study the generalization bound for the covariate shift problem, i.e., $p_T(x) \neq p_S(x)$ but $p_T(y|x) = p_S(y|x)$, where $p_S$ is the source distribution and $p_T$ is the target distribution. In contrast, Azizzadenesheli et al. (2019) provide a generalization bound for the label shift problem, i.e., $p_T(y) \neq p_S(y)$ but $p_T(x|y) = p_S(x|y)$.

**Domain adaptation**  While the literature on the domain adaptation problem is vast, we cover the most closely related works to ours here. A common method for the domain adaptation problem is to align the marginal distribution of the representation between the source and target domains. DANN (Ganin et al., 2016) employs a domain discriminator to classify the domain of a representation and maximizes its adversarial loss (a minimax game). WD (Shen et al., 2018) uses a neural network function $f$ (which is 1-Lipschitz continuous) to calculate the Wasserstein distance between two distributions and minimizes it. This is also a minimax game since the Wasserstein distance is the supremum over the search space of $f$. MMD (Gretton et al., 2012; Li et al., 2018) uses the maximum mean discrepancy to align the representation distribution. This method does not need a minimax objective; however, it requires the additional computation of several Gaussian kernels. Finally, CORAL (Sun and Saenko, 2016) matches the first two moments of the distribution; and while being a simple method, it fails to align more complex distributions. We consider these marginal alignment techniques our main baselines since our method falls into this category, and investigate the effectiveness of the reverse KL divergence in aligning the distributions of representation. Recently, more sophisticated alignment methods (Kang et al., 2019; Xu et al., 2019; Zhu et al., 2020) have been proposed for the domain adaptation problem, which achieve state-of-the-art performance. Instead of simply aligning the marginal distribution of the representation, these methods minimize the intra-class distance of the representation across domains, and possibly maximize the inter-class distance between them, using the MMD or L2 distance. However, they require pseudo labels for the target domain (often obtained via clustering). Moreover, they are complementary to our method, as we conjecture that our method can also be used in conjunction with these, leading to the same algorithms but with the KL distance instead of MMD or L2.

## 3  APPROACH

### 3.1  PROBLEM STATEMENT

In this paper, we consider one of the most common domain adaptation settings, which consists of a single-source domain $S$ with the joint data distribution $p_S(x, y)$ and a target domain $T$ with the data distribution $p_T(x, y)$, where $x$ denotes the input sample and $y$ is the label. We assume that these two domains have the same support sets $\mathcal{X}, \mathcal{Y}$. Regarding the training process of the domain adaptation problem, we further denote a labeled dataset of size $N_S$ sampled from the source domain $(x_S^{(i)}, y_S^{(i)})_{i=1}^{N_S}$, where $(x_S^{(i)}, y_S^{(i)}) \sim p_S(x, y)$, and an unlabeled dataset of size $N_T$ from the target domain $(x_T^{(i)})_{i=1}^{N_T}$, where $x_T^{(i)} \sim p_T(x)$.

The goal of a typical domain adaptation framework is to train a model with the labeled dataset of the source domain together with the unlabeled dataset from the target domain, so that the model will

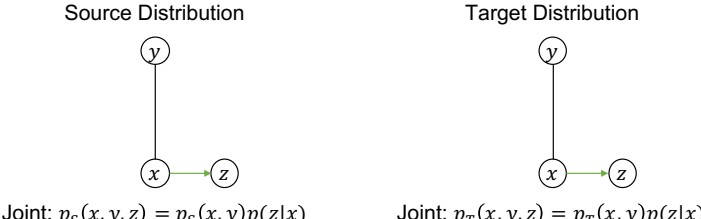

Figure 1: **Graphical model**. Note that the distribution $p(z|x)$ (green edge), corresponding to our representation network, is shared between the source and target domains.

perform decently in the target domain. Note that this is only effective if the labeling mechanism is not too different between the source and the target domains (Ben-David et al., 2010).

In the domain adaption problem, we expect the changes in the marginal distribution so that $p_S(x) \neq p_T(x)$, or the conditional distribution so that $p_S(y|x) \neq p_T(y|x)$, or both, which often render the typical empirical risk minimization training procedure ineffective. This motivates a line of approaches that learn a representation $z$ of $x$ whose marginal and conditional distributions are more aligned across the domains and use it for the prediction task, aiming at a better generalization performance to the target domain.

The general representation learning framework aims to learn a representation $z$ from $x$ with the mapping $p(z|x)$, which can be deterministic or probabilistic. That latent representation $z$ is expected to contain the label-related information; and is then used to predict the label $y$ (by a classifier). Note that since the source and target domains have the same support set for $x$ and share the representation mapping $p(z|x)$, they also have the same support set for $z$, denoted by $\mathcal{Z}$. Given the representation $z$, we learn a classifier to predict $y$ through the predictive distribution $\hat{p}(y|z)$ that is an approximation of the ground truth conditional distribution $p_S(y|z)$. During training, the representation network $p(z|x)$ and the classifier $\hat{p}(y|z)$ are trained jointly on the source domain and we "hope" that they can generalize to the target domain, meaning that both $p(z|x)$ and $\hat{p}(y|z)$ are kept unchanged between training and testing. The graphical model of that representation learning process is represented in Figure 1. In this paper, we consider a probabilistic representation mapping; specifically, the representation network will output $\mu(x)$ and $\sigma^2(x)$ and $p(z|x) = \mathcal{N}(z; \mu(x), \text{diag}(\sigma^2(x)))$, where $\mathcal{N}$ denotes a Gaussian distribution. This can also be thought of as a generalization of a deterministic representation, as we recover the deterministic case if $\sigma^2(x) \to 0$. Also, note that our method is not limited to the choice of this representation distribution, i.e., the discussion in this section holds for virtually any other distribution.

The joint distributions of $x, y, z$ for the source and target domains can be represented as follows

$$p_S(x, y, z) = p_S(x, y)p(z|x) \,, \qquad p_T(x, y, z) = p_T(x, y)p(z|x) \,. \tag{1}$$

and we define the predictive distribution of $y$ given $x$ as

$$\hat{p}(y|x) = \mathbb{E}_{p(z|x)}[\hat{p}(y|z)] \,. \tag{2}$$

**Remark 1.** *On the inference complexity of a probabilistic representation.*

Using a probabilistic representation, we need to sample multiple $z$ from $p(z|x)$ to estimate Eq. 2 during test time. However, this is not a big issue for the representation learning framework, since we only need to run the representation network $p(z|x)$ (which is usually deep) once to get a distribution of $z$. After sampling multiple $z$ from that distribution, we only need to rerun the classifier $\hat{p}(y|z)$, which is usually a small network (e.g., often contains one layer). Furthermore, we can also run $\hat{p}(y|z)$ (a small network) in parallel for multiple $z$ to reduce inference time if necessary.

During training, we usually sample a single $z$ from $p(z|x)$ for each $x$. The training objective is

$$l_{train} = \mathbb{E}_{x,y \sim p_S(x,y), z \sim p(z|x)}[-\log \hat{p}(y|z)] \tag{3}$$

$$\text{(this is also the upper bound of } \mathbb{E}_{p_S(x,y)}[-\log \hat{p}(y|x)] \text{ via Jensen Inequality)}$$

$$= \mathbb{E}_{p_S(z,y)}[-\log \hat{p}(y|z)] \tag{4}$$

where $-\log \hat{p}(y|z)$ is the loss of a "data point" $(z, y)$. For common choices of the predictive distribution in the classification and regression problems, this is a non-negative quantity. For example, for a classification problem with a categorical predictive distribution, this becomes the cross-entropy

loss, while for a regression problem with a Gaussian predictive distribution (with a fixed variance), it becomes the squared error (with an additive constant).

Minimizing $l_{train}$ will enforce $\hat{p}(y|z) \approx p_S(y|z)$.

We consider the below two assumptions of the representation $z$ on the source domain:

**Assumption 1.** *$I_S(z, y) = I_S(x, y)$, where $I_S(\cdot, \cdot)$ is the mutual information term, calculated on the source domain. In particular:*

$$I_S(z, y) = \mathbb{E}_{p_S(z,y)}\left[\log \frac{p_S(z, y)}{p_S(z)p_S(y)}\right]; \quad I_S(x, y) = \mathbb{E}_{p_S(x,y)}\left[\log \frac{p_S(x, y)}{p_S(x)p_S(y)}\right] \tag{5}$$

This is often referred to as the "sufficiency assumption" since it indicates that the representation $z$ has the same information about the label $y$ as the original input $x$, and is sufficient for this prediction task (in the source domain). Note that the data processing inequality indicates that $I_S(z, y) \leq I_S(x, y)$, so here we assume that $z$ contains maximum information about $y$.

**Remark 2.** *Assumption 1 is an optimization goal of the training process on the source domain.*

In particular, $l_{train}$ (with an additive constant) is an upper bound of $-I_S(z, y)$, which is an upper bound of $-I_S(x, y)$. Thus, minimizing $l_{train}$ will enforce $I_S(z, y)$ to be equal to $I_S(x, y)$. For a more detailed discussion of this, please refer to, for example, Alemi et al. (2016).

**Assumption 2.** $p_S(y|x) = \mathbb{E}_{p(z|x)}[p_S(y|z)] \quad \forall x, y \in \mathcal{X}, \mathcal{Y}$

When this assumption holds, the predictive distribution in Eq. 2 will approximate $p_S(y|x)$, as long as $\hat{p}(y|z)$ approximates $p_S(y|z)$.

**Remark 3.** *Assumption 2 is also an optimization goal of the training process on the source domain.*

This is because $l_{train}$ is an upper bound of $\mathbb{E}_{p_S(x,y)}[-\log \hat{p}(y|x)]$, which is an upper bound of $\mathbb{E}_{p_S(x,y)}[-\log p_S(y|x)]$. Thus, minimizing $l_{train}$ will enforce $\hat{p}(y|x)$ to be equal to $p_S(y|x)$. Therefore, $p_S(y|x) \approx \hat{p}(y|x) = \mathbb{E}_{p(z|x)}[\hat{p}(y|z)] \approx \mathbb{E}_{p(z|x)}[p_S(y|z)]$.

These two assumptions ensure that our network has good performance on the **source** domain. Note also that we only make the above two assumptions about the source domain, where we can enforce them through the training process. We do not make these assumptions about the target domain, since we have no access to the full target distribution. These two assumptions will also be used to prove our later theoretical result (Proposition 2).

## 3.2 KL GUIDED DOMAIN ADAPTATION

Now we will consider the test loss in the domain adaptation problem, and how we can reduce it. The test loss (of the target domain) is:

$$l_{test} = \mathbb{E}_{p_T(x,y)}[-\log \hat{p}(y|x)] = \mathbb{E}_{p_T(x,y)}[-\log \mathbb{E}_{p(z|x)}[\hat{p}(y|z)]] \tag{6}$$

$$\leq \mathbb{E}_{p_T(x,y)}[\mathbb{E}_{p(z|x)}[-\log \hat{p}(y|z)]] \quad \text{(Jensen Inequality)} \tag{7}$$

$$= \mathbb{E}_{p_T(z,y)}[-\log \hat{p}(y|z)] \tag{8}$$

Note that if the representation $z$ is invariant (both marginally and conditionally), then $p_T(z, y) = p_S(z, y)$ and Eq. 8 becomes $l_{train}$, and we have a perfect generalization between the source domain and the target domain. However, there is no way to guarantee the invariance, since we do not know the target domain and the target data distribution. In that case, we introduce the following proposition that ensures a generalization bound of the test loss based on the training loss and the KL divergence:

**Proposition 1.** *If the loss $-\log \hat{p}(y|z)$ is bounded by $M$ [1] $\forall z \in \mathcal{Z}, y \in \mathcal{Y}$, we have:*

$$l_{test} \leq l_{train} + \frac{M}{\sqrt{2}}\sqrt{\mathrm{KL}[p_T(y, z)|p_S(y, z)]} \tag{9}$$

$$= l_{train} + \frac{M}{\sqrt{2}}\sqrt{\mathrm{KL}[p_T(z)|p_S(z)] + \mathbb{E}_{p_T(z)}[\mathrm{KL}[p_T(y|z)|p_S(y|z)]]} \tag{10}$$

---

[1] In the classification problem, we can enforce this quite easily by augmenting the output softmax of the classifier so that each class probability is always at least $\exp(-M)$. For example, if we choose $M = 3 \Rightarrow \exp(-M) \approx 0.05$, and if the output softmax is $(p_1, p_2, ..., p_C)$, we can augment it into $(p_1 \cdot K + 0.05, p_2 \cdot K + 0.05, ..., p_C \cdot K + 0.05)$, where $K = 1 - 0.05 \cdot C$ and $C$ is the number of classes. This ensures the bound for the loss of a datapoint, while remaining the output prediction class.

*Proof.* provided in the appendix. □

This bound is similar to other bounds in the literature (e.g., Ben-David et al. (2010)) in the sense that it also contains the training loss, a marginal misalignment term and a conditional misalignment term ($\text{KL}[p_T(z)|p_S(z)]$ and $\mathbb{E}_{p_T(z)}[\text{KL}[p_T(z)|p_S(z)]]$ respectively in our case). However, Ben-David et al. (2010) consider a binary classification problem and their bounds are only practical for a deterministic labeling function (requires knowing the true labeling function, which is unknown for a probabilistic labeling mechanism, to compute the bound); while our bound works for the general case of supervised learning with any labeling mechanism. For a brief review of these bounds and a detailed discussion about their differences to ours, please refer to the appendix. Note that the bound in Proposition 1 is also true when applying to the input space directly (e.g., replacing $z$ with $x$). However, we are more interested in the bound in the representation space, since we can reduce it by regularizing the KL term.

To reduce the generalization gap, we want $p_T(z, y)$ to be close to $p_S(z, y)$. Aligning the marginal distribution (i.e., $p_S(z) \approx p_T(z)$) helps the classifier network $\hat{p}(y|z)$ avoid out-of-distribution data since the target representations it faces at test time belong to the source representation distribution which it was trained on; while aligning the conditional distribution ($p_S(y|z) \approx p_T(y|z)$) makes sure the classifier gives more accurate predictions on the target domain since $\hat{p}(y|z)$ was trained to approximate $p_S(y|z)$. In the domain adaptation problem, since we only have the unlabeled data from the target domain, we often align the marginal distribution of $z$ only. However, one problem is that the conditional misalignment also depends on the representation $z$, and when learning a representation $z$ that aligns the marginal, we might accidentally increase $\mathbb{E}_{p_T(z)}[\text{KL}[p_T(y|z)|p_S(y|z)]]$ at the same time, leading to a net increase in the above generalization bound. For example, what if (and is it possible that) the conditional misalignment increases to infinity while we learn a representation $z$?

Therefore, it is crucial that we can bound the above conditional misalignment. The below proposition handles this problem.

**Proposition 2.** *If Assumption 1 and 2 hold, and if $\frac{p_T(x,y)}{p_S(x,y)} < \infty$ (i.e., there exists N, which can be arbitrarily large, such that $\frac{p_T(x,y)}{p_S(x,y)} < N \; \forall x \in \mathcal{X}, y \in \mathcal{Y}$), we have:*

$$\mathbb{E}_{p_T(z)}[\text{KL}[p_T(y|z)|p_S(y|z)]] \leq \mathbb{E}_{p_T(x)}[\text{KL}[p_T(y|x)|p_S(y|x)]] \tag{11}$$

*Proof.* provided in the appendix. □

This shows that the conditional misalignment in the representation space is bounded by the conditional misalignment in the input space. This can also be viewed as an improvement over the analyses of Ben-David et al. (2010), where it is not clear if the conditional misalignment in the representation space is bounded or not. It then follows that:

$$l_{test} \leq l_{train} + \frac{M}{\sqrt{2}} \sqrt{\text{KL}[p_T(z)|p_S(z)] + \mathbb{E}_{p_T(x)}[\text{KL}[p_T(y|x)|p_S(y|x)]]}. \tag{12}$$

As mentioned earlier, in order for domain adaptation to be effective, we should expect that the labeling mechanism does not change too much (Ben-David et al., 2010). Thus, the conditional misalignment $\mathbb{E}_{p_T(x)}[\text{KL}[p_T(y|x)|p_S(y|x)]]$ is often small (and fixed – not dependent on the representation $z$). Therefore, to reduce the generalization bound, we can focus on minimizing $\text{KL}[p_T(z)|p_S(z)]$, with the objective:

$$l_{train} + \beta \text{KL}[p_T(z)|p_S(z)] \tag{13}$$

where $\beta$ is a hyper-parameter.

**Discussion on the use of reverse KL:** Our derivation leads to the reverse KL term $\text{KL}[p_T(z)|p_S(z)]$ as a regularizer of the distance between the two domains representations. We argue that there are several reasons that make this a good choice as a divergence between the source and target representation distributions. **(1)** First of all, as mentioned earlier, the KL term can be computed easily without any additional network or a minimax objective (details in Subsection 3.3). This leads to an efficient and stable training procedure, which often results in improved performance. **(2)** Secondly, the reverse KL has the zero-forcing/mode-seeking effect (Minka et al., 2005) that helps to alleviate the out-of-distribution problem. Specifically, the reverse KL forces the target representation distribution to have zero mass wherever the source distribution has zero mass (zero-forcing), thus

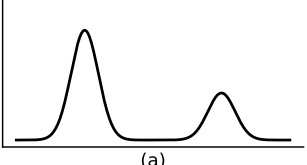 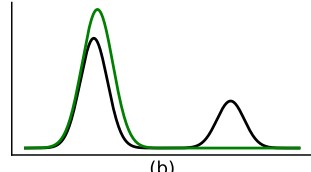 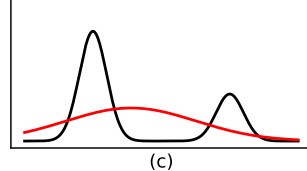

Figure 2: **Reverse KL** allows a flexible alignment of the representation while still effectively preventing the out-of-distribution problem. **(a)** Source representation distribution (black). Consider the case where the data distribution $p_S(x)$ of the source domain has two modes, then the representation distribution $p_S(z)$ will likely also have two modes; and consider the case where the target distribution has only one mode. **(b)** An acceptable target representation distribution (green) that helps the classifier avoid the out-of-distribution problem. Reverse KL allows for this type of flexible alignment (match to one/some of the modes) due to its mode-seeking nature. **(c)** A problematic target representation distribution (red), since the classification network will face out-of-distribution data at test time, in the area between the two modes. Reverse KL will prevent this due to its zero-forcing nature.

preventing the out-of-distribution data at test time (Figure 2c). On the other hand, its mode-seeking nature allows flexible alignment of the representation. For example, consider the case where the source domain is a mixture of two components (Figure 2a, i.e., it has two modes), and the target distribution is close to one of the two components. Ideally, we want to learn a representation network that matches the representation of the target domain to that of the corresponding component on the source mixture (Figure 2b). This representation will still perform well at test time since we would not have the out-of-distribution problem (the classification network is already trained on this mode of the source distribution). This flexible alignment (to one or some of the modes) is accepted by the reverse KL since it leads to a relatively small reverse KL value. Meanwhile, other methods such as DANN, MMD, CORAL and WD aim to match the representation distribution of the target domain and that of the whole source domain together, which could compress the representation too much, negatively affecting its expressive power. For instance, in the above example, trying to match the whole distribution of source and target domains based on other distance metrics might force the two modes of the source domain to collapse. The flexible alignment of the reverse KL (while still being very effective to prevent out-of-distribution data) might be beneficial in some practical cases.

We also empirically found that adding an auxiliary term $\mathrm{KL}[p_S(z)|p_T(z)]$ (forward KL) with a small coefficient $\beta_{aux}$ to the objective can help to align the distribution faster, leading to the objective:

$$l_{train} + \beta\mathrm{KL}[p_T(z)|p_S(z)] + \beta_{aux}\mathrm{KL}[p_S(z)|p_T(z)] \tag{14}$$

In practice, setting $\beta_{aux}$ to a (very) small value or zero often leads to the best results (using larger $\beta_{aux}$ can hurt the performance). Note that this does not invalidate our earlier discussion, as with such small values of $\beta_{aux}$, the alignment behavior is still dominated by the reverse KL.

### 3.3 OPTIMIZATION

In practice, we estimate Eq. 14 using minibatches. In particular, given a labelled minibatch $(\tilde{x}_S^{(i)}, \tilde{y}_S^{(i)})_{i=1}^B$ of the source domain and an unlabelled one $(\tilde{x}_T^{(i)})_{i=1}^B$ of the target domain, and a single sampled representation for each $x$: $(\tilde{z}_S^{(i)})_{i=1}^B$ and $(\tilde{z}_T^{(i)})_{i=1}^B$, we can get an unbiased estimator of the objective 14 as follows:

$$
\begin{aligned}
& l_{train} + \beta\mathrm{KL}[p_T(z)|p_S(z)] + \beta_{aux}\mathrm{KL}[p_S(z)|p_T(z)] \\
=& \mathbb{E}_{p_S(z,y)}[-\log\hat{p}(y|z)] + \beta\mathbb{E}_{p_T(z)}[\log p_T(z) - \log p_S(z)] + \beta_{aux}\mathbb{E}_{p_S(z)}[\log p_S(z) - \log p_T(z)] \\
\approx& \frac{1}{B}\sum_{i=1}^B -\log\hat{p}(\tilde{y}_S^{(i)}|\tilde{z}_S^{(i)}) + \beta\frac{1}{B}\sum_{i=1}^B\left[\log p_T(\tilde{z}_T^{(i)}) - \log p_S(\tilde{z}_T^{(i)})\right] \\
& + \beta_{aux}\frac{1}{B}\sum_{i=1}^B\left[\log p_S(\tilde{z}_S^{(i)}) - \log p_T(\tilde{z}_S^{(i)})\right]
\end{aligned}
\tag{15}
$$

However, it still requires knowing $p_S(z)$ and $p_T(z)$ to compute Eq. 15. We also use the minibatch to approximate these quantities:

$$p_S(z) = \mathbb{E}_{p_S(x)}[p(z|x)] \approx \frac{1}{B}\sum_{i=1}^{B} p(z|x_S^{(i)}); \quad p_T(z) = \mathbb{E}_{p_T(x)}[p(z|x)] \approx \frac{1}{B}\sum_{i=1}^{B} p(z|x_T^{(i)}). \quad (16)$$

Intuitively, we use a minibatch of data to construct a distribution of the representation $z$ (which is a mixture of $B$ components), and match that distribution for the two domains with the KL divergence. As mentioned earlier, we use a Gaussian distribution with a diagonal covariance matrix for $p(z|x)$ in practice, and employ the reparameterization trick (Kingma and Welling, 2013) to sample $z$.

Although the estimator in Eq. 15 is unbiased, the approximations in Eq. 16 will introduce some bias into our estimator (however, the estimator is still consistent, i.e., it becomes exact when $B \to \infty$). Therefore, the batch size might have an effect on the performance of the model. However, via an ablation study, we found that the effect of this bias estimator is not severe in practice, and our model achieves good performance even with a batch size of 64. For detailed results and discussion of this ablation study, please refer to the appendix C.4.

## 4 EXPERIMENTS

### 4.1 DATASETS

**RotatedMNIST** consists of 70,000 MNIST (LeCun et al., 2010) images that are divided into six domains, each with 11,666 images. The images in each domain are rotated counter-clockwise by $0°, 15°, 30°, 45°, 60°$ and $75°$ respectively. We denote the six domains as $\mathcal{M}_0, \mathcal{M}_{15}, \mathcal{M}_{30}, \mathcal{M}_{45}, \mathcal{M}_{60}$ and $\mathcal{M}_{75}$. We use $\mathcal{M}_0$ as the source domain, and perform five experiments, each with $\mathcal{M}_{15}, \mathcal{M}_{30}, \mathcal{M}_{45}, \mathcal{M}_{60}$ or $\mathcal{M}_{75}$ as the target domain. The task is classification of the ten digit labels.

**DIGITS** is a common domain adaptation dataset, with 3 digit classification sub-datasets, namely MNIST, USPS (Hull, 1994) and SVHN (Netzer et al., 2011). Three common adaptation experiments are MNIST $\to$ USPS, USPS $\to$ MNIST and SVHN $\to$ MNIST.

**VisDA17** (Peng et al., 2017) is a challenging real-world classification dataset with a simulation-to-real adaptation task. This dataset contains over 280K images from 12 classes. The source domain contains renderings of 3D models, while the target domain contains real images.

Please refer to the appendix for results of datasets with more domains such as **PACS (Li et al., 2017)**.

### 4.2 BASELINES

We consider all common marginal alignment methods for domain adaptation as our baselines, including **DANN**, **MMD**, **CORAL** and **WD**. We also consider **ERM** (empirical risk minimization) and its variant **ERM (prob)** (same as ERM but with the probabilistic representation network used in our model). For ERM, DANN, MMD and CORAL, we follow the implementation by Gulrajani and Lopez-Paz (2020); while for ERM (prob) and WD, we use our own implementation in Pytorch (Paszke et al., 2019). Note that we do not include methods that are not from the marginal-alignment family since they are out of the scope of this paper. For the full description of these baselines, please refer to our appendix and the official code at https://github.com/atuannguyen/KL.

### 4.3 EXPERIMENTAL SETTING

For the RotatedMNIST and DIGITS experiments, we use a simple convolutional neural network with four $3 \times 3$ convolutional layers (followed by an average pooling layer) as the representation network. For VisDA17, we use a Resnet50 as the representation network. Only the last layer of the representation network differs for a deterministic representation (ERM, DANN, CORAL, MMD, WD) and a probabilistic one (ERM (prob) and KL (ours)). For a representation of size $d_z$, the last layer's dimension of a deterministic representation network is $d_z$, while that of a probabilistic network is $2 \cdot d_z$ ($d_z$ for $\mu$ and $d_z$ for $\sigma^2$). Please refer to the appendix for the detailed experimental setting (including data split, hyper-parameter tuning for each model, evaluation protocol, etc.)

Table 1: Rotated MNIST experiments with $\mathcal{M}_0$ as the source domain.

| Model | Target Domain | | | | | |
|---|---|---|---|---|---|---|
| | $\mathcal{M}_{15}$ | $\mathcal{M}_{30}$ | $\mathcal{M}_{45}$ | $\mathcal{M}_{60}$ | $\mathcal{M}_{75}$ | Average |
| ERM | 97.5±0.2 | 84.1±0.8 | 53.9±0.7 | 34.2±0.4 | 22.3±0.5 | 58.4 |
| ERM (prob) | 96.8±0.3 | 83.2±1.6 | 51.3±0.9 | 31.4±1.1 | 20.7±0.7 | 56.7 |
| DANN | 97.3±0.4 | 90.6±1.1 | 68.7±4.2 | 30.8±0.6 | 19.0±0.6 | 61.3 |
| MMD | 97.5±0.1 | 95.3±0.4 | 73.6±2.1 | 44.2±1.8 | 32.1±2.1 | 68.6 |
| CORAL | 97.1±0.3 | 82.3±0.3 | 56.0±2.4 | 30.8±0.2 | 27.1±1.7 | 58.7 |
| WD | 96.7±0.3 | 93.1±1.2 | 64.1±3.3 | 41.4±7.6 | 27.6±2.0 | 64.6 |
| KL (ours) | **97.8±0.1** | **97.1±0.2** | **93.4±0.8** | **75.5±2.4** | **68.1±1.8** | **86.4** |

Table 2: DIGITS and VisDA17 experiments.

| Model | DIGITS | | | | VisDA17 |
|---|---|---|---|---|---|
| | $M \rightarrow U$ | $U \rightarrow M$ | $S \rightarrow M$ | Average | $S \rightarrow R$ |
| ERM | 73.1±4.2 | 54.8±6.2 | 65.9±1.4 | 64.6 | 39.1±0.5 |
| ERM (prob) | 70.3±3.2 | 59.0±8.3 | 67.6±1.3 | 65.6 | 37.2±2.2 |
| DANN | 90.7±0.4 | 91.2±0.8 | 71.1±0.5 | 84.3 | 57.7±1.3 |
| MMD | 91.8±0.3 | 94.4±0.5 | 82.8±0.3 | 89.7 | 62.8±1.1 |
| CORAL | 88.0±1.9 | 83.3±0.1 | 69.3±0.6 | 80.2 | 39.5±4.5 |
| WD | 88.2±0.6 | 60.2±1.8 | 68.4±2.5 | 72.3 | 38.9±4.8 |
| KL (ours) | **98.2±0.2** | **97.3±0.5** | **92.5±0.9** | **96.0** | **70.6±0.5** |

## 4.4 RESULTS

**RotatedMNIST and DIGITS:** Table 1 and Table 2 show the results for the RotatedMNIST and DIGITS experiments. It is clear that in these experiments, aligning the representation between domains does help improve the generalization performance. Among the baselines (DANN, MMD, CORAL, WD), MMD performs the best, which we attribute to the fact that it does not use a minimax objective, leading to more stable optimization. Meanwhile, CORAL performs the worst, since it only matches the first two moments of the distributions and might fail to align complex distributions. Our method, KL, largely outperforms the baselines, indicating its effectiveness. Visualization of the representation space also shows that our method aligns the representation better than existing methods. This visualization can be found in the appendix.

**VisDA17** (Table 2): In this challenging dataset, many marginal alignment techniques (CORAL, WD) fail the adaptation task (achieve similar accuracy as the ERM baselines). MMD and KL (ours) are again the best performers, confirming that a stable training objective is beneficial in practice. Our method outperforms all other marginal-alignment approaches significantly, suggesting the effectiveness of the KL divergence in representation alignment.

## 5 CONCLUSION

In conclusion, in this paper, we derive a generalization bound of the target loss in the domain adaptation problem using the reverse KL divergence. We then show that with a probabilistic representation, the KL divergence can easily be estimated using Monte Carlo (minibatch) samples, without any additional computation or adversarial objective. By minimizing the KL divergence, we can reduce the generalization bound and have a better guarantee about the test loss. We also empirically show that our method outperforms relevant baselines with large margins, which we attribute to its simple and stable training procedure and the mode-seeking/zero-forcing nature of the reverse KL. We conclude that KL divergence is very effective as a tool for representation alignment. In general, a limitation of marginal alignment methods (ours included) is that when the conditional distribution changes significantly from the source domain to the target domain, aligning the marginal would not help the target domain's performance. This is also reflected in our generalization bound. For future work, we would want to investigate the use of KL divergence in other types of alignment. For example, we can follow the algorithm in Kang et al. (2019) to minimize the intra-class distance of the representation across domains and maximize the inter-class distance between them, but using the KL divergence instead of MMD as the distance between representation distributions. Another direction would be using KL divergence to align the conditional distribution across domains in a multi-source setting.

**Acknowledgments** This work is supported by the UKRI grant: Turing AI Fellowship EP/W002981/1 and EPSRC/MURI grant: EP/N019474/1. We would also like to thank the Royal Academy of Engineering and FiveAI.

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

# A    PROOFS

For the following proofs, we treat the variables as continuous variables and always use the integral. If one or some of the variables are discrete, it is straight-forward to replace the corresponding integral(s) with summation sign(s) and the proofs still hold.

## A.1    PROPOSITION 1

*Proof.* We have:

$$l_{test} \leq \mathbb{E}_{p_T(z,y)}[-\log \hat{p}(y|z)] \tag{17}$$

$$= \int -\log \hat{p}(y|z) p_T(z,y) dz dy \tag{18}$$

$$= \int -\log \hat{p}(y|z) p_S(z,y) dz dy + \int -\log \hat{p}(y|z) \left[ p_T(z,y) - p_S(z,y) \right] dz dy \tag{19}$$

$$= l_{train} + \int -\log \hat{p}(y|z) \left[ p_T(z,y) - p_S(z,y) \right] dz dy \tag{20}$$

Let $\mathcal{A} = \{(z,y)|p_T(z,y) - p_S(z,y) \geq 0\}$ and $\mathcal{B} = \{(z,y)|p_T(z,y) - p_S(z,y) < 0\}$, using the fact that $-\log \hat{p}(y|z) \geq 0 \ \forall z \in \mathcal{Z}, y \in \mathcal{Y}$ we have:

$$\int -\log \hat{p}(y|z) \left[ p_T(z,y) - p_S(z,y) \right] dz dy \tag{21}$$

$$= \int_{\mathcal{A}} -\log \hat{p}(y|z) \left[ p_T(z,y) - p_S(z,y) \right] dz dy + \int_{\mathcal{B}} -\log \hat{p}(y|z) \left[ p_T(z,y) - p_S(z,y) \right] dz dy \tag{22}$$

$$\leq \int_{\mathcal{A}} -\log \hat{p}(y|z) \left[ p_T(z,y) - p_S(z,y) \right] dz dy \tag{23}$$

$$= \int_{\mathcal{A}} -\log \hat{p}(y|z) \left| p_T(z,y) - p_S(z,y) \right| dz dy \tag{24}$$

$$\leq M \int_{\mathcal{A}} \left| p_T(z,y) - p_S(z,y) \right| dz dy \tag{25}$$

$$(\text{since} -\log \hat{p}(y|z) \leq M) \tag{26}$$

where $|.|$ is the absolute value.

Here, $\int_{\mathcal{A}} |p_T(z,y) - p_S(z,y)| \, dz dy$ is also called the total variation of the two distributions $p_T(z,y)$ and $p_S(z,y)$.

Note that:

$$\int p_T(z,y) - p_S(z,y)dzdy = 0 \tag{27}$$

$$\Leftrightarrow \int_{\mathcal{A}} p_T(z,y) - p_S(z,y)dzdy + \int_{\mathcal{B}} p_T(z,y) - p_S(z,y)dzdy = 0 \tag{28}$$

$$\Leftrightarrow \int_{\mathcal{A}} p_T(z,y) - p_S(z,y)dzdy = \int_{\mathcal{B}} p_S(z,y) - p_T(z,y)dzdy \tag{29}$$

$$\Leftrightarrow \int_{\mathcal{A}} |p_T(z,y) - p_S(z,y)|\, dzdy = \int_{\mathcal{B}} |p_T(z,y) - p_S(z,y)|\, dzdy \tag{30}$$

$$\Leftrightarrow \int_{\mathcal{A}} |p_T(z,y) - p_S(z,y)|\, dzdy = \frac{1}{2}\int |p_T(z,y) - p_S(z,y)|\, dzdy \tag{31}$$

Therefore:

$$l_{test} \le l_{train} + M \int_{\mathcal{A}} |p_T(z,y) - p_S(z,y)|\, dzdy \tag{32}$$

$$= l_{train} + \frac{M}{2}\int |p_T(z,y) - p_S(z,y)|\, dzdy \tag{33}$$

Using the Pinsker's inequality, we have:

$$\left(\int |p_T(z,y) - p_S(z,y)|\, dzdy\right)^2 \le 2\int p_T(z,y)\log\frac{p_T(z,y)}{p_S(z,y)}dzdy \tag{34}$$

Therefore, we finally have:

$$l_{test} \le l_{train} + \frac{M}{2}\sqrt{2\int p_T(z,y)\log\frac{p_T(z,y)}{p_S(z,y)}dzdy} \tag{35}$$

$$= l_{train} + \frac{M}{\sqrt{2}}\sqrt{\mathrm{KL}[p_T(z,y)|p_S(z,y)]} \tag{36}$$

Which concludes our proof.

Also note that the KL divergence between $p_T(z,y)$ and $p_S(z,y)$ can further be decomposed into the marginal misalignment and conditional misalignment as follow:

$$\mathrm{KL}[p_T(z,y)|p_S(z,y)] = \mathbb{E}_{p_T(z,y)}[\log p_T(z,y) - \log p_S(z,y)] \tag{37}$$

$$= \mathbb{E}_{p_T(z,y)}[\log p_T(z) + \log p_T(y|z) - \log p_S(z) - \log p_S(y|z)] \tag{38}$$

$$= \mathbb{E}_{p_T(z,y)}[\log p_T(z) - \log p_S(z)] + \mathbb{E}_{p_T(z,y)}[\log p_T(y|z) - \log p_S(y|z)] \tag{39}$$

$$= \mathbb{E}_{p_T(z)}[\log p_T(z) - \log p_S(z)]$$
$$+ \mathbb{E}_{p_T(z)}\left[\mathbb{E}_{p_T(y|z)}[\log p_T(y|z) - \log p_S(y|z)]\right] \tag{40}$$

$$= \mathrm{KL}[p_T(z)|p_S(z)] + \mathbb{E}_{p_T(z)}\left[\mathrm{KL}[p_T(y|z)|p_S(y|z)]\right] \tag{41}$$

$$\square$$

### A.2 PROPOSITION 2

*Proof.* According to Assumption 1, we have:

$$I_S(z,y) = I_S(x,y) \tag{42}$$

$$\Leftrightarrow H_S(y) - H_S(y|z) = H_S(y) - H_S(y|x) \tag{43}$$

$$\Leftrightarrow H_S(y|z) = H_S(y|x) \tag{44}$$

$$\Leftrightarrow \mathbb{E}_{p_S(z,y)}[\log p_S(y|z)] = \mathbb{E}_{p_S(x,y)}[\log p_S(y|x)] \tag{45}$$

$$\Leftrightarrow \mathbb{E}_{p_S(x,z,y)}[\log p_S(y|z)] = \mathbb{E}_{p_S(x,y)}[\log p_S(y|x)] \tag{46}$$

$$\Leftrightarrow \mathbb{E}_{p_S(x,y)}\left[\mathbb{E}_{p(z|x)}[\log p_S(y|z)]\right] = \mathbb{E}_{p_S(x,y)}[\log p_S(y|x)] \tag{47}$$

$$\Leftrightarrow \mathbb{E}_{p_S(x,y)}\left[\log p_S(y|x) - \mathbb{E}_{p(z|x)}[\log p_S(y|z)]\right] = 0 \tag{48}$$

According to Assumption 2, $\forall x \in \mathcal{X}, y \in \mathcal{Y}$ we have:

$$p_S(y|x) = \mathbb{E}_{p(z|x)}[p_S(y|z)] \tag{49}$$

$$\Leftrightarrow \log p_S(y|x) = \log \mathbb{E}_{p(z|x)}[p_S(y|z)] \tag{50}$$

$$\Rightarrow \log p_S(y|x) \geq \mathbb{E}_{p(z|x)}[\log p_S(y|z)] \tag{51}$$

Since $\frac{p_T(x,y)}{p_S(x,y)} < \infty$, there exists $N > 0$ such that $\frac{p_T(x,y)}{p_S(x,y)} \leq N \; \forall x \in \mathcal{X}, y \in \mathcal{Y}$. Therefore:

$$\mathbb{E}_{p_T(x,y)}\left[\log p_S(y|x) - \mathbb{E}_{p(z|x)}[\log p_S(y|z)]\right] \tag{52}$$

$$= \mathbb{E}_{p_S(x,y)}\left[\left(\log p_S(y|x) - \mathbb{E}_{p(z|x)}[\log p_S(y|z)]\right)\frac{p_T(x,y)}{p_S(x,y)}\right] \tag{53}$$

$$\leq N.\mathbb{E}_{p_S(x,y)}\left[\log p_S(y|x) - \mathbb{E}_{p(z|x)}[\log p_S(y|z)]\right] \tag{54}$$

$$= 0 \tag{55}$$

Therefore:

$$\mathbb{E}_{p_T(x,y)}\left[\log p_S(y|x) - \mathbb{E}_{p(z|x)}[\log p_S(y|z)]\right] = 0 \tag{56}$$

$$\Leftrightarrow \mathbb{E}_{p_T(x,y)}\left[\log p_S(y|x)\right] = \mathbb{E}_{p_T(x,y,z)}[\log p_S(y|z)] \tag{57}$$

$$\Leftrightarrow \mathbb{E}_{p_T(x,y)}\left[\log p_S(y|x)\right] = \mathbb{E}_{p_T(z,y)}[\log p_S(y|z)] \tag{58}$$

We have:

$$\mathbb{E}_{p_T(z)}\left[\mathrm{KL}[p_T(y|z)|p_S(y|z)]\right] \leq \mathbb{E}_{p_T(x)}\left[\mathrm{KL}[p_T(y|x)|p_S(y|x)]\right] \tag{59}$$

$$\Leftrightarrow \mathbb{E}_{p_T(z,y)}\left[\log p_T(y|z) - \log p_S(y|z)\right] \leq \mathbb{E}_{p_T(x,y)}\left[\log p_T(y|x) - \log p_S(y|x)\right] \tag{60}$$

Using Eq 58, we now only need to prove that:

$$\mathbb{E}_{p_T(z,y)}\left[\log p_T(y|z)\right] \leq \mathbb{E}_{p_T(x,y)}\left[\log p_T(y|x)\right] \tag{61}$$

$$\Leftrightarrow -H_T(y|z) \leq -H_T(y|x) \tag{62}$$

$$\Leftrightarrow H_T(y) - H_T(y|z) \leq H_T(y) - H_T(y|x) \tag{63}$$

$$\Leftrightarrow I_T(z,y) \leq I_T(x,y) \tag{64}$$

$$\text{(always true based on the Data Processing Inequality)} \tag{65}$$

$$\square$$

## B    REVIEW OF EXISTING GENERALIZATION BOUNDS

There have been several works studying the generalization bounds of the Domain Adaptation problem. We briefly review the most important and common ones here with a discussion about their differences to our proposed bound.

### B.1    BEN-DAVID ET AL. (2010)

Ben-David et al. (2010) consider a binary classification problem. Let $x$ be the input with the support set $\mathcal{X}$ and $y$ be the binary label with the support set $\mathcal{Y} = \{0, 1\}$. Consider a source domain with a distribution $P_X^s$ over the input $x$ and the true labeling function $f^s : \mathcal{X} \to \{0, 1\}$; and similarly a target domain with a distribution $P_X^t$ over the input $x$ and the true labeling function $f^t : \mathcal{X} \to \{0, 1\}$. Note that the authors claim that this labeling function can be probabilistic; in that case, $f : \mathcal{X} \to [0, 1]$ denoting the probability. However, we argue that this probabilistic setting is impractical since we would not know that true underlying function in order to calculate/estimate the bounds in practice). Therefore, we found that the bound is only practical for the case of a deterministic labeling mechanism.

The error of the classifier $h$, which is also a deterministc labeling function, on the source domain is:

$$\epsilon^s(h) = \mathbb{E}_{x \sim P_X^s}[|h(x) - f^s(x)|], \tag{66}$$

and similarly for the target domain:

$$\epsilon^t(h) = \mathbb{E}_{x \sim P_X^t}[|h(x) - f^t(x)|]. \tag{67}$$

Here $|.|$ is the absolute value, which means the loss of a data point is the L1 distance of the labels.

Consider a hypothesis space $\mathcal{H}$ and let a classifier $h$ be any function from that space. The first theorem in Ben-David et al. (2010) offers a bound of the target loss $\epsilon^t(h)$ based on the source loss $\epsilon^s(h)$, and the total variation between $P_X^s$ and $P_X^t$, and the difference between the two labeling function $f^s$ and $f^t$:

**Theorem 1 (Ben-David et al. (2010))**

$$\epsilon^t(h) \leq \epsilon^s(h) + 2d_1(P_X^s, P_X^t) + \min_{P_X \in \{P_X^s, P_X^t\}} \mathbb{E}_{x \sim P_X}[|f^s(x) - f^t(x)|] \tag{68}$$

where $d_1(P_X^s, P_X^t)$ is the total variational distance, i.e., $d_1(P_X^s, P_X^t) := \sup_{\mathcal{A} \in \mathscr{X}} [P_X^s(\mathcal{A}) - P_X^t(\mathcal{A})]$, and $\mathscr{X}$ is the sigma-field of $\mathcal{X}$ (set of all subsets of $\mathcal{X}$).

In this theorem, the term $2d_1(P_X^s, P_X^t)$ presents the marginal misalignment and $\min_{P_X \in \{P_X^s, P_X^t\}} \mathbb{E}_{x \sim P_X}[|f^s(x) - f^t(x)|]$ is the conditional misalignment.

Ben-David et al. (2010) also propose another bound based on a variant of the $\mathcal{H}$-divergence, which is presented in the following theorem:

**Theorem 2 (Ben-David et al. (2010))**

$$\epsilon^t(h) \leq \epsilon^s(h) + d_{\mathcal{H}\Delta\mathcal{H}}(P_X^s, P_X^t) + \lambda_{\mathcal{H}} \tag{69}$$

where the $\mathcal{H}\Delta\mathcal{H}$-divergence $d_{\mathcal{H}\Delta\mathcal{H}}(P_X^s, P_X^t) := \sup_{h_1, h_2 \in \mathcal{H}} |\Pr_{x \sim P_X^s}[h_1(x) \neq h_2(x)] - \Pr_{x \sim P_X^t}[h_1(x) \neq h_2(x)]|$ replaces the total variation to measure the marginal misalignment of the two domains. Meanwhile, $\lambda_{\mathcal{H}} = \inf_{h \in \mathcal{H}}[\epsilon^s(h) + \epsilon^t(h)]$ measures the conditional misalignment of the two domains (if the two true labeling functions $f^s$ and $f^t$ are the same and belong to the hypothesis space $\mathcal{H}$, this quantity is zero).

The above bounds can also be applied to the representation space (similar to ours), leading to the same bounds where the input $x$ is replaced by its representation $z$. However, as mentioned in the main text, it is not clear if the conditional misalignment in the representation space (e.g., $\min_{P_Z \in \{P_Z^s, P_Z^t\}} \mathbb{E}_{z \sim P_Z}[|g^s(z) - g^t(z)|]$) is bounded or not.

**Difference to our bound**    First of all, Ben-David et al. (2010) only consider a binary classification problem. Moreover, as discussed above, the bounds in Ben-David et al. (2010) are only practical with deterministic labeling mechanisms for both domains. This assumption is hard to be true for most datasets since the labeling mechanism is usually probabilistic. This makes it not generalizable to the general case of supervised learning. Furthermore, the loss function is a $L_1$ distance between the labeling function, which is also not a common choice in practice, which makes it challenging to generalize to the multiclass classification set-up (even if there exists a deterministic labeling function for a multiclass dataset, using the $L_1$ loss for the one-hot encoded labels would be unreasonable; the common loss function in practice for the multiclass classification problem is the cross-entropy loss). Finally, the total variation and $\mathcal{H}$-divergence might be hard to estimate in practice since it requires the computation of a supremum.

## B.2 MANSOUR ET AL. (2009)

Mansour et al. (2009) consider a more flexible problem set-up than Ben-David et al. (2010). Specifically, instead of $\mathcal{Y} = \{0, 1\}$, they consider the cases where $\mathcal{Y} = \{0, 1\}$ (for binary classification) or $\mathcal{Y}$ is a measurable subset of $\mathbb{R}$ (for regression). Note that their bound still cannot work for multiclass classfication. They also generalize the L1 loss function to a loss function $L : \mathcal{Y} \times \mathcal{Y} \to \mathbb{R}$; however, this loss function must obey the triangle inequality. Although the L1 distance satisfies this inequality, it is not generally true for other common loss functions in practice (e.g., cross-entropy). They still consider **deterministic** labeling function $f^s$ and $f^t$ for the source and target domain.

Similar to Ben-David et al. (2010), with a hypothesis $h$ from the hypothesis space $\mathcal{H}$, the error of the source and target domain are:

$$\epsilon^s(h) = \mathbb{E}_{x \sim P_X^s}[L(h(x), f^s(x))] \tag{70}$$

$$\epsilon^t(h) = \mathbb{E}_{x \sim P_X^t}[L(h(x), f^t(x))] \tag{71}$$

For convenience, denote also the error between two labeling function $h$ and $h'$ in the source and target distribution as:

$$\epsilon^s(h, h') = \mathbb{E}_{x \sim P_X^s}[L(h(x), h'(x))] \tag{72}$$

$$\epsilon^t(h, h') = \mathbb{E}_{x \sim P_X^t}[L(h(x), h'(x))] \tag{73}$$

(which means $\epsilon^s(h) = \epsilon^s(h, f^s)$ and $\epsilon^t(h) = \epsilon^t(h, f^t)$).

Also, let $h^{*s}$ and $h^{*t}$ be the minimizer of $\epsilon^s(h)$ and $\epsilon^t(h)$ respectively. In particular:

$$h^{*s} = \arg\min_{h \in \mathcal{H}} \epsilon^s(h) = \arg\min_{h \in \mathcal{H}} \mathbb{E}_{x \sim P_X^s}[L(h(x), f^s(x))] \tag{74}$$

$$h^{*t} = \arg\min_{h \in \mathcal{H}} \epsilon^t(h) = \arg\min_{h \in \mathcal{H}} \mathbb{E}_{x \sim P_X^t}[L(h(x), f^t(x))] \tag{75}$$

Mansour et al. (2009) introduce a generalization bound as follow:

**Theorem 3 (Mansour et al. (2009))** Assume that the loss function $L$ is symmetric and obeys the triangle inequality. Then, for any hypothesis $h \in \mathcal{H}$, the following holds

$$\epsilon^t(h) \le \epsilon^t(h^{*t}) + \epsilon^s(h, h^{*s}) + \text{disc}(P_X^s, P_X^t) + \epsilon^t(h^{*s}, h^{*t}) \tag{76}$$

where $\text{disc}(P_X^s, P_X^t) := \sup_{h, h' \in \mathcal{H}} |\epsilon^s(h, h') - \epsilon^t(h, h')|$, which is a generalized version of the $\mathcal{H}\Delta\mathcal{H}$-divergence.

Here, the first term $\epsilon^t(h^{*t})$ is the ideal target loss (will be zero if the hypothesis space $\mathcal{H}$ contains $f_t$), the second term $\epsilon^s(h, h^{*s})$ will be zero if we choose $h = h^{*s}$ (which is the common practice, e.g., train the classifier $h$ on the source domain), the third term $\text{disc}(P_X^s, P_X^t)$ measures the marginal misalignment, and the final term $\epsilon^t(h^{*s}, h^{*t})$ is somewhat an indicator of the conditional misalignment (becomes zero if $f^s = f^t \in \mathcal{H}$).

**Difference to our bound**  The above bound is based on the ideal target loss, while our bound is based on the source loss. In practice, we have (an estimate) of the source loss calculated on the source domain's training set; meanwhile, the ideal target loss is unknown. This makes the above bound less useful in practice compared to ours. Furthermore, the above bound has similar problems as the ones in Ben-David et al. (2010): it does not work for multiclass classification, it assumes a deterministic labeling mechanism (which does not hold in practice), it assumes the loss function obeys the triangle inequality (which generally is not true in practice), and it contains terms that are not easy to compute in practice (supremum and infimum).

## C  ADDITIONAL EXPERIMENTAL RESULTS

### C.1  VISUALIZATION OF THE ROTATEDMNIST EXPERIMENTS

Figure 3 shows the representation space (for the RotatedMNIST experiment with source $\mathcal{M}_0$ and target $\mathcal{M}_{45}$) of our method compared to the baselines, visualized using t-SNE (Van der Maaten and Hinton, 2008). We can clearly see that the color clusters (which correspond to the digit classes) of our method are much more aligned when compared to other baselines such as MMD, DANN and ERM. This illustrates the effectiveness of our method in aligning the representation in the domain adaptation problem.

### C.2  PACS

Table 3 presents the results for PACS, which is a challenging real-world dataset for domain adaptation/generalization. In this dataset, our model outperforms the ERM baselines by roughly 9%

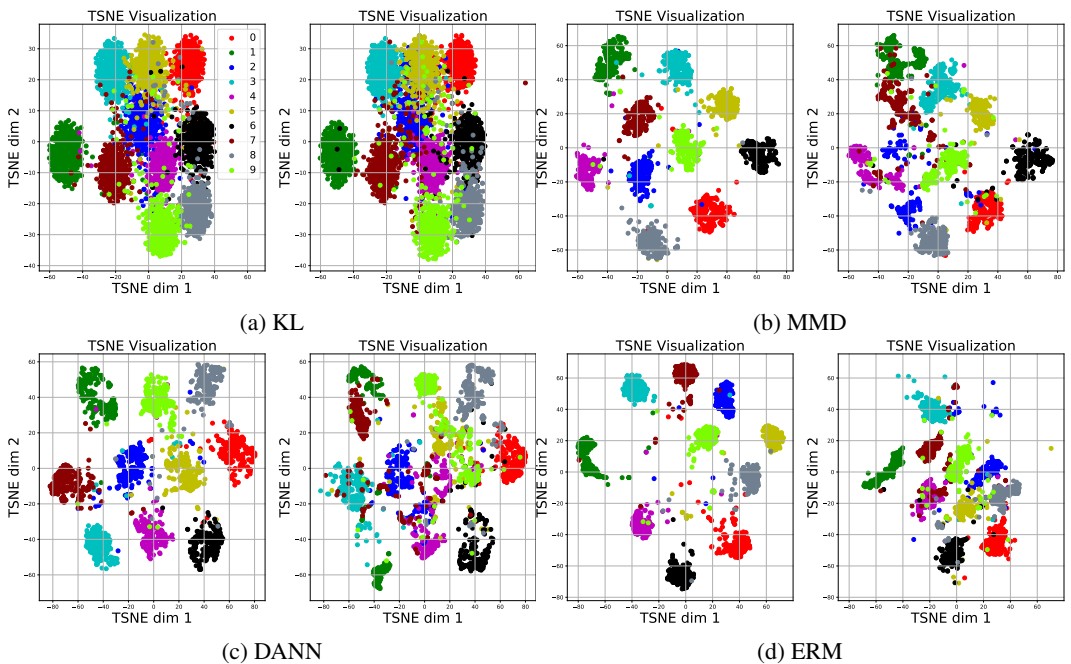

Figure 3: Visualization using t-SNE of the representation space of our method KL and the baselines MMD, DANN, ERM. For each method, the left subfigure corresponds to the source domain $\mathcal{M}_0$ and the right one corresponds to the target domain $\mathcal{M}_{45}$. Each color represents a digit class.

on average, indicating the effectiveness of our representation-alignment technique. Our method is the best performer (with a large margin) on 8 out of 12 experiments, showing a clear benefit over other representation alignment techniques. Together with our method, MMD again performs the best among the representation-alignment baselines (DANN, MMD, CORAL and WD), confirming that a stable training procedure (with no minimax objectives in MMD and our model) is important and often leads to better results. It is also worth noting that our model still outperforms MMD despite being less computationally expensive (in this implementation, MMD needs to compute seven Gaussian kernels for each of three pairs of representation sets in each minibatch).

It is interesting that the ERM baselines perform the best in some experiments (e.g., S → C, S → P). This result also agrees with the one observed in Gulrajani and Lopez-Paz (2020) that domain generalization/adaptation techniques might have negative effects when applied unsuccessfully. It should be noted that the S (sketch) domain is undoubtedly the most different compared to others (only black sketch on a white background while other domains have colors), which might explain the difficulty when learning to transfer between domains.

### C.3    OFFICE-31

We also provide additional results on the Office-31 dataset Saenko et al. (2010). Table 4 presents the performance of our model and two of the best performing baselines, namely MMD and DANN. Although our method is not state-of-the-art in this dataset (and understandingly so), it significantly outperforms the relevant baselines considered in this paper.

### C.4    ABLATION STUDY: EFFECT OF BATCH SIXE

In this subsection, we conduct an ablation study to investigate the effect of the batch size on our model's performance. Table 5 shows the performance of our method on the RotatedMNIST dataset, with $\mathcal{M}_0$ as the source domain and $\mathcal{M}_{45}$ as the target domain and with various choices of the batch size. As expected, our model's performance tends to benefit from a bigger batch size, since it would alleviate the bias of our objective estimator. We therefore recommend increasing the batchsize whenever possible. However, our model performs well even for a batch size as small as 64 (which is considered small in this era of deep learning).

Table 3: PACS experiments.

| Experiments | Model | | | | | | |
| | ERM | ERM (prob) | DANN | MMD | CORAL | WD | KL (ours) |
|---|---|---|---|---|---|---|---|
| A → C | 66.1±1.3 | 63.5±0.8 | 71.0±3.2 | **79.5±0.4** | 62.7±10.4 | 76.2±0.9 | 73.1±3.4 |
| A → P | 94.3±0.6 | 93.5±1.3 | 94.5±0.5 | 94.5±1.1 | 86.3±6.8 | 92.4±1.3 | **95.4±1.2** |
| A → S | 53.6±0.8 | 60.9±3.5 | 58.6±12.8 | 62.1±2.0 | 46.2±3.5 | 53.9±2.7 | **67.4±1.9** |
| C → A | 69.7±1.1 | 70.8±2.3 | 76.4±1.7 | 79.5±3.0 | 75.9±0.9 | 69.0±2.1 | **83.3±1.1** |
| C → P | 82.0±0.9 | 81.5±2.1 | 78.6±3.4 | 80.8±2.3 | 78.3±3.6 | 72.9±8.6 | **83.1±7.4** |
| C → S | 72.2±1.4 | 70.4±1.5 | **76.1±1.0** | 74.1±1.3 | 56.9±11.0 | 48.7±6.1 | 68.2±0.5 |
| P → A | 65.7±2.3 | 63.3±1.2 | 68.0±2.7 | 67.7±1.8 | 70.0±1.5 | 62.6±1.5 | **75.5±2.5** |
| P → C | 29.1±1.9 | 27.2±3.3 | 50.7±5.0 | 47.4±0.8 | 47.5±8.6 | 56.1±1.4 | **67.7±1.2** |
| P → S | 38.0±1.0 | 35.9±2.3 | 29.3±9.8 | 59.7±4.8 | 15.8±5.3 | 22.3±15.0 | **64.5±2.1** |
| S → A | 41.3±6.5 | 40.9±3.9 | 39.2±3.5 | 40.0±3.3 | 39.1±4.8 | 36.1±9.5 | **48.2±2.4** |
| S → C | 66.7±1.0 | **67.9±1.4** | 64.3±2.0 | 65.7±2.3 | 59.9±1.5 | 60.5±2.0 | 63.5±0.4 |
| S → P | **49.3±3.3** | 46.0±4.7 | 44.3±4.0 | 45.1±0.9 | 37.4±2.7 | 38.5±5.6 | 39.1±3.4 |
| Average | 60.6 | 60.2 | 62.6 | 66.3 | 56.3 | 57.4 | **69.1** |

Table 4: Office-31 experiments

| Model | A→D | A→W | D→A | D→W | W→A | W→D | Average |
|---|---|---|---|---|---|---|---|
| DANN | 79.7±0.4 | 82.0±0.4 | 68.2±0.4 | 96.9±0.2 | 67.4±0.5 | 99.1±0.1 | 82.2 |
| MMD | 75.5±0.6 | 73.4±0.4 | 60.8±1.0 | 97.4±0.5 | 61.5±1.1 | 99.5±0.2 | 78.0 |
| KL | 85.6±0.6 | 87.9±0.4 | 70.1±1.1 | 99.0±0.2 | 69.3±0.7 | 100.0±0.0 | 85.3 |

Table 5: **Ablation study: Effect of batch size**. Rotated MNIST experiments with $\mathcal{M}_0$ source and $\mathcal{M}_{45}$ target.

| Batch size | 256 | 128 | 64 | 32 |
|---|---|---|---|---|
| KL (ours) | 93.4±0.8 | 93.4±1.2 | 93.3±0.3 | 89.5±0.9 |

Table 6: **Ablation study: Effect of auxiliary term**. RotatedMNIST.

| Model | $\mathcal{M}_{15}$ | $\mathcal{M}_{30}$ | $\mathcal{M}_{45}$ | $\mathcal{M}_{60}$ | $\mathcal{M}_{75}$ | Average |
|---|---|---|---|---|---|---|
| KL ($\beta_{aux} = 0$) | 97.8±0.5 | 96.6±0.4 | 92.0±0.4 | 68.8±4.6 | 62.3±4.2 | 83.5 |
| KL (reported in the paper) | 97.8±0.1 | 97.1±0.2 | 93.4±0.8 | 75.5±2.4 | 68.1±1.8 | 86.4 |

Table 7: **Ablation study: Effect of auxiliary term**. DIGITS and VisDA17.

| Model | DIGITS | | | | VisDA17 |
| | M → U | U → M | S → M | Average | S → R |
|---|---|---|---|---|---|
| KL ($\beta_{aux} = 0$) | 98.2±0.2 | 97.1±0.4 | 90.0±1.1 | 95.1 | 67.8 |
| KL (reported in the paper) | 98.2±0.2 | 97.3±0.52 | 92.5±0.9 | 96.0 | 70.5 |

## C.5 Ablation Study: Effect of the auxiliary forward KL

In this subsection, we investigate the contribution of the auxiliary forward KL term, by considering a variant of our method without this term ($\beta_{aux} = 0$). Table 6 and Table 7 show the results for this ablation experiment. Clearly, most of the improvement comes from the reverse KL regularizer term. And even without the auxiliary forward KL term, our results are still significantly higher than the baselines. Note that the auxiliary term requires virtually no extra computation, so we believe adding the auxiliary term is reasonable.

## D Detailed Experimental Settings

In each experiment, we split both the source and the target data into two portions: 80% and 20%. We use 80% of the source domain data and 80% of the target domain data (without the labels) as the training data. We use the remaining 20% of the source data as the validation set, and the remaining 20% of the target domain data as the test set. Note that we do not use the labeled data from the target domain during training or validation. This evaluation protocol is recommended by Gulrajani and Lopez-Paz (2020).

## D.1 BASELINES

**ERM** Bousquet et al. (2003) is the typical empirical risk minimization training procedure, meaning that the model is trained normally in the training data and does not account for the distribution shift (domain adaptation).

**ERM (prob):** since we use a probabilistic network, we also include the probabilistic version of ERM. This is similar to ERM but uses a probabilistic representation network (same as ours).

**DANN** Ganin et al. (2016) utilizes a discriminator to distinguish the representation from the source and target domains. It uses an adversarial loss to enforce that the distributions of the representation from the source domain and the target domain are the same.

**MMD** Li et al. (2018) uses the maximum mean discrepancy (MMD) to align the representation's distributions.

**CORAL** Sun and Saenko (2016) aligns the representation distributions of the source and target domains by matching their first two moments.

**WD** Shen et al. (2018) uses the Wasserstein distance to match the distribution of the representation.

## D.2 REPRESENTATION NETWORK USED IN ROTATEDMNIST AND DIGITS

We use a simple CNN as the representation network in this experiment. This network is exactly the same as the one used in Gulrajani and Lopez-Paz (2020).

Our code is in PyTorch. The network is constructed by the following layers, where `output_dim=256` for a deterministic representation network and `output_dim=512` for a probabilistic representation network (256 for $\mu$ and 256 for $\sigma^2$):

```
- Conv2d(in_channels=1,out_channels=64,kernel_size=3,stride=1,padding=1)
- ReLU()
- GroupNorm(num_groups=8,num_channels=64)
- Conv2d(in_channels=64,out_channels=128,kernel_size=3,stride=2,padding=1)
- ReLU()
- GroupNorm(num_groups=8,num_channels=128)
- Conv2d(in_channels=128,out_channels=128,kernel_size=3,stride=1,padding=1)
- ReLU()
- GroupNorm(num_groups=8,num_channels=128)
- Conv2d(in_channels=128,out_channels=output_dim,kernel_size=3,stride=1,padding=1)
- ReLU()
- GroupNorm(num_groups=8,num_channels=output_dim)
- AdaptiveAvgPool2d(output_size=(1,1))
```

## D.3 HYPER-PARAMETERS TUNING

We train each model for 100 epochs. To avoid hyperparameter bias, we tune the hyperparameters (learning rate, regularizer coefficients, weight decay, representation dimension and dropout rate) for each method and dataset independently. Following Gulrajani and Lopez-Paz (2020), we perform a random search (Bergstra and Bengio, 2012) to tune hyperparameters for the baselines. We use the Adam optimizer Kingma and Ba (2014) for all the models. We re-run each set of hyperparameters three times. We train all models on an NVIDIA Quadro RTX 6000 GPU.

The readers can also refer to our source code for the experiment setting.

Below are the hyper-parameters considered by the random search in our experiments for each baseline. {.} means a set of hyper-parameters considered, while [., .] means a range of hyper-parameters considered.

### D.3.1 ROTATEDMNIST AND DIGITS

The representation's dimension is 128 (details in Section D.2). For even more details about the below hyper-parameters, please refer to our provided code.

- **ERM**: learning rate: $[10^{-4.5}, 10^{-2.5}]$, weight decay: 0.0, dropout rate: 0.0, batch size: $[8, 512]$, number of layers of the classifier ($\hat{p}(y|z)$): 1 or 3.

- **ERM (prob)**: learning rate: $[10^{-4.5}, 10^{-2.5}]$, weight decay: 0.0, dropout rate: 0.0, batch size: $[8, 512]$, number of layers of the classifier ($\hat{p}(y|z)$): 1 or 3.

- **DANN Ganin et al. (2016)**: learning rate: $[10^{-4.5}, 10^{-2.5}]$, weight decay: 0.0, dropout rate: 0.0, batch size: $[8, 512]$, number of layers of the classifier ($\hat{p}(y|z)$): 1 or 3, adversarial loss coefficient: $[10^{-2}, 10^{2}]$, weight decay of discriminator: $[10^{-6}, 10^{-2}]$, number of discriminator steps per generator steps: $\{1, 2, 4, 8\}$, grad penalty coefficient: $[10^{-2}, 10^{1}]$

- **MMD Li et al. (2018)**: learning rate: $[10^{-4.5}, 10^{-2.5}]$, weight decay: 0.0, dropout rate: 0.0, batch size: $[8, 512]$, number of layers of the classifier ($\hat{p}(y|z)$): 1 or 3, MMD coefficient: $[10^{-3}, 10^{-1}]$

- **CORAL Sun and Saenko (2016)**: learning rate: $[10^{-4.5}, 10^{-2.5}]$, weight decay: 0.0, dropout rate: 0.0, batch size: $[8, 512]$, number of layers of the classifier ($\hat{p}(y|z)$): 1 or 3, CORAL loss coefficient: $[10^{-3}, 10^{-1}]$

- **WD Shen et al. (2018)**: learning rate: $[10^{-4.5}, 10^{-2.5}]$, weight decay: 0.0, dropout rate: 0.0, batch size: $[8, 512]$, number of layers of the classifier ($\hat{p}(y|z)$): 1 or 3, wasserstein distance coefficient: $[10^{-2}, 10^{2}]$, weight decay of network $f$: $[10^{-6}, 10^{-2}]$, number of $f$ optimization steps per normal optimization steps: $\{1, 2, 4, 8\}$, grad penalty coefficient: $[10^{-2}, 10^{1}]$

- **KL (ours)**: learning rate: $[10^{-4.5}, 10^{-2.5}]$, weight decay: 0.0, dropout rate: 0.0, batch size: 256, number of layers of the classifier ($\hat{p}(y|z)$): 1, $\beta : 0.3$, $\beta_{aux} : 0.1$

### D.3.2 VISDA17 AND PACS

The representation network for VisDA17 is a Resnet50, while the one for PACS is a Resnet18. For even more details about the below hyper-parameters, please refer to our provided code.

- **ERM**: learning rate: $[10^{-5}, 10^{-3.5}]$, weight decay: $[10^{-6}, 10^{-2}]$, dropout rate: $\{0.0, 0.1, 0.5\}$, batch size: $[8, 45]$, representation dimension: $\{16, 128, 256, 512\}$, number of layers of the classifier ($\hat{p}(y|z)$): 1 or 3.

- **ERM (prob)**: learning rate: $[10^{-5}, 10^{-3.5}]$, weight decay: $[10^{-6}, 10^{-2}]$, dropout rate: $\{0.0, 0.1, 0.5\}$, batch size: $[8, 45]$, representation dimension: $\{16, 128, 256, 512\}$, number of layers of the classifier ($\hat{p}(y|z)$): 1 or 3.

- **DANN Ganin et al. (2016)**: learning rate: $[10^{-5}, 10^{-3.5}]$, weight decay: $[10^{-6}, 10^{-2}]$, dropout rate: $\{0.0, 0.1, 0.5\}$, batch size: $[8, 45]$, representation dimension: $\{16, 128, 256, 512\}$, number of layers of the classifier ($\hat{p}(y|z)$): 1 or 3, adversarial loss coefficient: $[10^{-2}, 10^{2}]$, weight decay of discriminator: $[10^{-6}, 10^{-2}]$, number of discriminator steps per generator steps: $\{1, 2, 4, 8\}$, grad penalty coefficient: $[10^{-2}, 10^{1}]$

- **MMD Li et al. (2018)**: learning rate: $[10^{-5}, 10^{-3.5}]$, weight decay: $[10^{-6}, 10^{-2}]$, dropout rate: $\{0.0, 0.1, 0.5\}$, batch size: $[8, 45]$, representation dimension: $\{16, 128, 256, 512\}$, number of layers of the classifier ($\hat{p}(y|z)$): 1 or 3, MMD coefficient: $[10^{-3}, 10^{-1}]$

- **CORAL Sun and Saenko (2016)**: learning rate: $[10^{-5}, 10^{-3.5}]$, weight decay: $[10^{-6}, 10^{-2}]$, dropout rate: $\{0.0, 0.1, 0.5\}$, batch size: $[8, 45]$, representation dimension: $\{16, 128, 256, 512\}$, number of layers of the classifier ($\hat{p}(y|z)$): 1 or 3, CORAL loss coefficient: $[10^{-3}, 10^{-1}]$

- **WD Shen et al. (2018)**: learning rate: $[10^{-5}, 10^{-3.5}]$, weight decay: $[10^{-6}, 10^{-2}]$, dropout rate: $\{0.0, 0.1, 0.5\}$, batch size: $[8, 45]$, representation dimension: $\{16, 128, 256, 512\}$, number of layers of the classifier ($\hat{p}(y|z)$): 1 or 3, wasserstein distance coefficient: $[10^{-2}, 10^{2}]$, weight decay of network $f$: $[10^{-6}, 10^{-2}]$, number of $f$ optimization steps per normal optimization steps: $\{1, 2, 4, 8\}$, grad penalty coefficient: $[10^{-2}, 10^{1}]$

- **KL (ours)**: learning rate: $10^{-4}$, weight decay: $[10^{-6}, 10^{-2}]$, dropout rate: 0.0, batch size: 256, representation dimension: 16, number of layers of the classifier ($\hat{p}(y|z)$): 1, $\beta : \{0.1, 0.05, 0.001\}$, $\beta_{aux} : \{0.1, 0.05, 0.01, 0.0\}$

