# OpenReview forum: "KL Guided Domain Adaptation"
_ICLR.cc/2022/Conference — ICLR 2022 Poster_

### Official Review · Reviewer_V8Y3 · 2021-11-01

**Correctness:** 3
**Technical Novelty And Significance:** 2
**Empirical Novelty And Significance:** 3
**Recommendation:** 6
**Confidence:** 3

**Main Review:**

Strength:
1. the idea is simple and the paper is easy to follow.
2. the experimental result looks good.

weakness:
1. My most concerning part is the change from equation (12) to equation (13). Although KL(p_T(y|x) | p_S(y|x)) is fixed over the whole dataset. But in training, we sample a small mini-batch for training, and this term could be changed and unstable.
2. Experiments are weak since all datasets except VisDA-17 are not popular domain adaptation benchmarks. Why not perform experiments on Office-31 and Office-Home datasets? Also, the results in this paper on VisDA-18 are not consistent with existing work such as [Zhu'20]. In addition, [Zhu'20] is based on marginal MMD and you need to compare it.
3. the novelty. I am not familiar with domain adaption, but the proposed idea in this paper is very simple and straightforward. So I actually don’t believe there are no previous works that also tried KL divergence for representation alignment (for example, I searched “KL divergence domain adaption” and got many results).  So what is the new in this paper? The KL divergence over probabilistic representation?

[Zhu'20] Zhu Y, Zhuang F, Wang J, et al. Deep subdomain adaptation network for image classification[J]. IEEE transactions on neural networks and learning systems, 2020, 32(4): 1713-1722.

**Summary Of The Paper:**

this paper proposed a method for domain adaption.
The idea and method are quite simple. Especially, KL divergence on representation alignment is used. And the experimental result looks promising on the selected task.

**Summary Of The Review:**

The paper is easy to follow, and the idea is easy and effective. However, some assumptions seem too strong and some datasets are missed in the experiment part.

---

> ### Author Response · Authors · 2021-11-22
> **Rebuttal**
>
> Dear reviewer,
>
> Please see our answers and clarification below:
>
> **My most concerning part is the change from equation (12) to equation (13). Although KL(p_T(y|x) | p_S(y|x)) is fixed over the whole dataset. But in training, we sample a small mini-batch for training, and this term could be changed and unstable.**
>
> First of all, let us make a small clarification. The term $\mathbb{E}_{p_T(x)}[\text{KL}[p_T(y|x)|p_T(y|z)]]$ is fixed over the **true underlying data distributions** $p_S(x,y)$ and $p_T(x,y)$. Nevertheless, it is indeed true that using the whole datasets to estimate this would be pretty accurate (given that we somehow know $p_T(y|x)$ and $p_S(y|x)$).
>
> However, our argument is not quite related to this. We are interested in minimizing $l_{test}$, which is an expectation over the **true underlying** target data distribution $p_T(x,y)$ (not an estimation using full batch or minibatch). This target loss is bounded by 3 terms as in the RHS of Eq.12, all of which are expectations over the data distribution(s). Since the last term is fixed (only depends on the two data distribution $p_S(x,y)$ and $p_T(x,y)$, but not the representation being learned), to reduce this bound, we only need to reduce the first two terms (hence the objective function in Eq.13, note that the two terms here are still expectations over the true data distributions).
>
> Then, from Eq.13 and Eq.14, we use a minibatch to get an estimator for the objective (including the above two terms and an auxiliary term), leading to Eq.15. This is similar to using a minibatch to estimate an objective function in Stochastic Gradient Descent. Note that as in SGD, in the long run, the variance of the estimator will cancel out. Again, we are not (and don't need to) minimizing the term $\mathbb{E}_{p_T(x)}[\text{KL}[p_T(y|x)|p_T(y|z)]]$ here.
>
> Please let us know if you would like to have further discussion regarding this point.
>
> **Experiments are weak since all datasets except VisDA-17 are not popular domain adaptation benchmarks. Why not perform experiments on Office-31 and Office-Home datasets?**
>
> To the best of our knowledge, both the DIGITS and VisDA17 are very common benchmarks in the domain adaptation. However, thanks to your suggestion (as well as reviewer FBaS), we are conducting extra experiments (the Office-31 dataset as you both suggested). We will update the result in a separate comment.
>
> **Also, the results in this paper on VisDA-17 are not consistent with existing work such as [Zhu'20].**
>
> We recheck the results of VisDA-17 from [Zhu'20] and found that their result matches our number. The only model (among the baselines considered in our paper) that [Zhu'20] reported is DANN, which they reported to achieve 57.6% accuracy. The number reported in our paper is 57.7.

---

> > ### Author Response · Authors · 2021-11-22
> > **Rebuttal (continued)**
> >
> > **In addition, [Zhu'20] is based on marginal MMD and you need to compare it.**
> >
> > We believe this baseline is out of the scope of our paper. Specifically, we aim to improve the marginal alignment technique (which is the backbone in most DA algorithms, including [Zhu'20]), thus we only consider marginal alignment methods as our baselines. [Zhu'20] proposed a method to align the representation distribution within each class (some refer to as the reverse conditional distribution) using MMD. This is not a direct baseline to ours since the same algorithm can be used but with the KL divergence instead of the MMD distance. In this sense, if KL divergence is a good criterion to align the distribution (compared to MMD and other baselines listed in our paper), this has great potential to improve more complex alignment algorithms such as [Zhu'20], and would be a promising future research direction. This exact point has been discussed in our related work section, and we quote it here "Recently, more sophisticated alignment methods (Kang et al., 2019; Xu et al., 2019) have been proposed for the domain adaptation problem, which achieve state-of-the-art performance. Instead of simply aligning the marginal distribution of the representation, these methods minimize the intra-class distance of
> > the representation across domains, and maximize the inter-class distance between them, using the MMD or L2 distance [...] they are complementary to our method, as we conjecture that our method can also be used in conjunction with these, leading to the same algorithms but with the KL distance instead of MMD or L2."
> >
> > We will also add [Zhu'20] into this discussion point of related work.
> >
> > **the novelty. I am not familiar with domain adaption, but the proposed idea in this paper is very simple and straightforward. So I actually don’t believe there are no previous works that also tried KL divergence for representation alignment (for example, I searched “KL divergence domain adaption” and got many results). So what is the new in this paper? The KL divergence over probabilistic representation?**
> >
> > There are some works that propose bounds for the domain adaptation problem using the f-divergence family (including the KL divergence). However, the difference and the contributions of our work are: (1) To the best of our knowledge, we are the first to investigate the effectiveness of using KL divergence for marginal alignment in domain adaptation. (2) We propose a simple way to estimate this KL term in practice and show that the resulting method is very effective. (3) On the theoretical side, we also show that under some reasonable assumptions, the conditional misalignment in the representation space is bounded by the conditional misalignment in the input space, which is an improvement over existing analyses and makes the marginal alignment technique sound. [4] Our setting is more general: our bound works for all commonly-used loss functions, works for a probabilistic labeling mechanism without knowing the true labeling function, works for all cases of supervised learning.
> >
> > Best regards,
> >
> > Authors.

---

> > > ### Comment · Reviewer_V8Y3 · 2021-11-30
> > > **Thank you for the response**
> > >
> > > Thank you for the detailed response. I will boost the score to 6.

---

> > > > ### Author Response · Authors · 2021-11-30
> > > > **Thank you**
> > > >
> > > > Thank you for your reconsideration!
> > > >
> > > > Best regards,
> > > >
> > > > Authors.

---

### Official Review · Reviewer_g5v8 · 2021-11-02

**Correctness:** 4
**Technical Novelty And Significance:** 3
**Empirical Novelty And Significance:** 3
**Recommendation:** 8
**Confidence:** 4

**Main Review:**

- The paper is well written and easy to follow. The problem is well-motivated, and the derived bound and optimization procedure seem to be reasonable.
- What I don't see convincing is the jump and oversimplification in Eq(15) where the last line replaces the expectation with the simple sum assuming that each of p_S(z) and p_T(z) to be uniformly distributed. This issue is neither mentioned nor discussed.
- Similar to the previous comment, Eq(16) assumes that $p_S(x)$ and p_T(x) to be uniform!
- I am wondering why the authors show selectivity when it comes to the results on the digits data sets. Why don't you show the different combinations? U-->S, S-->U, M U-->S.
-Finally, the results in Table 1 show a clear failure of all the baselines on M75 and M60 *(achieving half of what KL achieves). I find it difficult to accept these results.


**Summary Of The Paper:**

The paper proposes a domain adaptation method that is guided by the reverse Kullback-Leibler (KL) divergence between the target and source domains in the representation space. To this end, it is assumed that the representation z is sufficient and that p_S(y|x) = E_p(z|x)[p_S(y|z)].
The authors derive a bound on the test loss based on the training loss and a quantity based on the divergence of the joint distribution (y,z) between the target and source domain (Proposition 1). Since this bound requires labels from both domains, Proposition 2 helps to use the divergence computation on p_S(y|x) instead of p_S(y|z). The proposed method and optimization procedure are evaluated on multiple data sets and compared with multiple baselines.


**Summary Of The Review:**

All-in-all, I find the paper of a great potential contribution to the field. I only have two issues (i) the sudden strange simplifications in equations 15 and 16, and (ii) selective experiments besides the weird results in Table 1.

Update:

I read the other reviews and the authors' responses, I'm fine with the authors' reply. So I am keeping my score.

---

> ### Author Response · Authors · 2021-11-19
> **Rebuttal**
>
> Dear reviewer,
>
> Thank you for your time and effort in helping review our paper. Please see our detailed justification/clarification below:
>
> **What I don't see convincing is the jump and oversimplification in Eq(15) where the last line replaces the expectation with the simple sum assuming that each of p_S(z) and p_T(z) to be uniformly distributed. This issue is neither mentioned nor discussed.**
>
> We believe there is a slight misunderstanding here. We are using the Monte Carlo sampling, that for **any** distrubution $p(a)$ of a random variable $a$ (not necessary uniform) and function $f(a)$, then an unbiased estimator of $\mathbb{E}_{p(a)}[f(a)]$ is $\frac{1}{k}(f(a_1)+f(a_2)+...+f(a_k))$, with $a_1,a_2,...,a_k$ sampled from the distribution $p(a)$.
>
> We will make this clearer in the paper.
>
> **Similar to the previous comment, Eq(16) assumes that p_S(x) and p_T(x) to be uniform!**
>
> Similarly, we are using unbiased estimators via Monte Carlo sampling as above.
>
>
> **I am wondering why the authors show selectivity when it comes to the results on the digits data sets. Why don't you show the different combinations? U-->S, S-->U, M U-->S.**
>
> We were following the standard experimental setting as the domain adaptation, for example [1,2]. To the best of our knowledge, the adaptation task U-->S, S-->U, M U --> S is not very popular in the literature. Please let us know if you would like us to run extra experiments for these less popular tasks.
>
> [1] Long, Mingsheng, et al. "Conditional adversarial domain adaptation." arXiv preprint arXiv:1705.10667 (2017).
> [2] Saito, Kuniaki, et al. "Maximum classifier discrepancy for unsupervised domain adaptation." Proceedings of the IEEE conference on computer vision and pattern recognition. 2018.
>
> **Finally, the results in Table 1 show a clear failure of all the baselines on M75 and M60 (achieving half of what KL achieves). I find it difficult to accept these results.**
>
> We were also surprised by how other marginal alignment methods fail on domains M60 and M75 of this dataset. However, we have checked the results carefully. We also provide the source code if any of the readers want to run the experiments themselves. Note that we tune the hyperparameters independently for each baseline and each dataset (with grid search), so hyperparameter tuning is not a problem here.
>
> Best regards,
>
> Authors.

---

### Official Review · Reviewer_FBaS · 2021-11-02

**Correctness:** 3
**Technical Novelty And Significance:** 2
**Empirical Novelty And Significance:** 2
**Recommendation:** 3
**Confidence:** 4

**Main Review:**

**Strengths.**

- The idea of a probabilitisc interpretation of DA is interesting.
- The resulting algorithm seems  simple to implement in practice.

**Weakness:**

- **Major Concern #1** Novelty of the theoretical result.

The paper mentioned several times (e.g. first paragraph of related work) that existing theory of domain adaptation e.g. [1] is limited to the case of binary labels. While the results shown  in [1] are indeed for the binary classification setting,  I do not necesarily agree this  limitation exists, in [1] this was done for simplicity as it is typical, the extension to the multiclass setting just follows. Moreover, the multi-class setting using a novel discrepancy measure that extends [1] was analyzed in [2].  In [3] the authors  further extend the results from [1] to general f-divergences and also derived rigorous generalization bounds. The family of f-divergences include KL and r KL as particular cases.  A lot of work has also be done since [1] (2010) for example [1,2,3,4] and comparison, improvement wrt to them is not discussed .

Moreover, if the goal is to show a bound where the discrepancy between marginals is estimated using the KL divergence rather than L1 (as in [1])  that could be  obtained by directly applying Pinsker's inequality on top of [1] Theorem 1.  How would that compare to the current setting?

- **Major Concern #2** Significance of the derived bound.

One of the major motivation for the H-divergence introduced in [1] vs L1 and then several authors using hypot-based divergences such as [2,3,4]. It is that (quoting [1]) "L1 is an overly strict measure that unnecessarily inflates the bound". Pinsker's inequality shows KL  upper bounds  L1. Wouldnt the direct use of the KL then inflates more the bound? .  How does writing a generalization bound using the KL divergence which is further ubounded solves this problem?


- **Major Concern #3** Comparison of the algorithm with modern baselines. The latest baseline WD is from 2018. Why there is no comparison to recent work in domain-adaptation? I would also recommend the authors to do experiments in the office-31 dataset as it has become standard practice. This will also make the comparison vs existing work easier.

- **Major Concern #4** No experimental analysis is provided.

[1] A theory of learning from different domains. Springer 2010

[2] Bridging theory and algorithm for domain adaptation. ICML 2019

[3] f-Domain Adversarial Learning: Theory and Algorithms. ICML 2021

[4] On learning invariant representations for domain adaptation,ICML 2020

[5] Office-31 Dataset: [https://www.cc.gatech.edu/~judy/domainadapt/](https://www.cc.gatech.edu/~judy/domainadapt/)

**Summary Of The Paper:**

 In this paper, the authors propose a generalization bound for domain adaptation where discrepancy between distributions (i.e. marginals and class conditionals) is estimated using the reverse KL divergence. Inspired by this result, the authors then proposed a learning algorithm and show  that this outperforms some existing methods.

**Summary Of The Review:**

Overall, I consider this work to have some interesting ideas however I consider it needs a major rewriting and clarity on the contribution of the theoretical results. In my opinion, they look more as a way to motivate an algorithm which could be done following the results from either [1,2,3,4] rather than a new generalization bound as claimed. The writing should also be more careful when comparing and criticizing previous work. I also believe comparison to recent DA algorithms/methods is need it, Similarly, more experimental analysis in order to validate the effectivity of the proposed algorithm is also need it.

---

> ### Author Response · Authors · 2021-11-22
> **Rebuttal**
>
> Dear reviewer,
>
> Thank you for helping review and improve our paper. Please see our detailed answers below:
>
> **Novelty of the theoretical result.**
>
> Admittedly, we miss the work of [3]. However, the differences of our theoretical result compared to [3] are:
>
> - The loss function in [3] needs to satisfy the triangle inequality, which is not true for most loss functions used in the literature such as the cross-entropy loss. Also, the loss functions considered in [1] and [2] are also unconventional.
> - [3] needs to know the true labeling function (optimal Bayes classifier) for a probabilistic labeling mechanism to calculate the loss term. This is not available in practice. Also, common loss functions used in practice (e.g., cross-entropy, squared error) only takes the instance label and the prediction as input, not the true labeling function.
> - We are able to prove that the conditional misalignment in the representation space is bounded by the conditional misalignment in the input space (Proposition 2), while [3] wasn't able to show a similar result. Indeed, they explicitly assume that the ideal joint risk in the representation is negligible. This assumption might be too strong because we don't know the representation distribution (and more importantly it is not fixed).
>
> **Moreover, if the goal is to show a bound where the discrepancy between marginals is estimated using the KL divergence rather than L1 (as in [1]) that could be obtained by directly applying Pinsker's inequality on top of [1] Theorem 1. How would that compare to the current setting?**
>
> This is not correct. Our setting is much more general than [1]. For example, we don't need to know the true labeling function to calculate the loss. A detailed comparison between our work and [1] is provided in the related work section and also in the appendix. Furthermore, as mentioned above, an important theoretical result of our paper is also Proposition 2.
>
> **Significance of the derived bound.**
>
> It is indeed true that using Pinsker's Inequality inflates the bound a bit; however, we argue that:
> - The inflated amount is only significant when the distribution discrepancy is not regularized. When the discrepancy (KL) is minimized like in our case, both the terms (total variation and KL) is quite close to 0, so the inflation is not significant. Therefore, it only comes down to which distance/divergence is most efficient/effective to align the distribution. As we argue in the paper, the reverse KL has several properties that are suitable for the alignment problem in domain adaptation.
> - Moreover, we also have other theoretical contribution (other than the above bound), namely Proposition 2.
>
> **Comparison of the algorithm with modern baselines. Office31 benchmark.**
>
> As the aim of our paper is to improve marginal alignment technique (as this is an important backbone for more advanced DA methods), we only compare against plain marginal alignment methods (align the representation distribution of two domains). We will post the results of Office31 in a separate comment.
>
> **No experimental analysis is provided.**
>
> We did provide some analysis on the effect of the batch size on our model's performance. Due to the page limit, we move this analysis to Appendix C.2. We will also add an ablation study of the auxiliary term as suggested by reviewer i12Z to our revised version. If you would like to see any other analyses, please let us know.
>
> Best regards,
>
> Authors.

---

> ### Comment · Reviewer_FBaS · 2021-11-23
> **Response**
>
> I thank the authors for the response and the updated revision.
> After reading the response and also other reviewers' concerns, several of my concerns still remain and therefore I have decided to keep my score.
>
> __wrt to theoretical results.__ I think there might be an overseeing of the difference between a generalization bound (and the purpose of this) vs a theoretically motivated algorithm.  I believe this paper proposes a method for the latter, while the presentation suggests the former. Several of the bounds in DA [1,2,3,4,6] have been derived using hypothesis-based discrepancies because directly using divergences or distance functions (such as l1) could make the bound vacuous even in favorable adaptation situations [6]. The KL upper bounds L1, which already unnecessarily inflates the bound [1]. Using the proposed results,  it may be worth considering the simple example where the two distributions ps and pt are simple Gaussians with \mu_2= \mu_1+4 (and/or vice-versa). Likely, and independently of the complexity of the function class the divergence term with KL may dominate(>1). In conclusion, while  I agree the losses in the generalization bounds from  [1,2,3,6] are not necessarily the ones used by the algorithm in practice, they are a good proxy and that does not invalidate the results. Besides, assumptions such as triangle inequality and access to the true labeling function might have been taken for simplicity. And this might not be as critical as measuring the dissimilarity between ps and pt directly using a divergence, which is further unbounded.
>
> __wrt to experimental results and comparison vs modern baselines.__ Several recent works on DA can be understood as “plain marginal alignment methods” and those could also be used as a backbone for more advanced DA methods. For example, why there is no comparison vs [2] 2019?. This is necessary in order to show the effectiveness of the proposed algorithm.
>
> __wrt to additional experimental analysis.__ See an example of analysis about alignment in the paper of the baselines used and literature referenced.
>
>
> Overall, I like the ideas presented in this work and consider some are very interesting, however, I believe the work is not ready. It needs a major revision and restructure on the contribution of the theoretical results. Moreover, a comparison of the proposed algorithm vs recent works is needed to show the experimental significance of the proposed algorithm and for it to be ready for publication.
>
> ---
> [1] A theory of learning from different domains. Springer 2010
>
> [2] Bridging theory and algorithm for domain adaptation. ICML 2019
>
> [3] f-Domain Adversarial Learning: Theory and Algorithms. ICML 2021
>
> [4] On learning invariant representations for domain adaptation, ICML 2020
>
> [5] Office-31 Dataset: https://www.cc.gatech.edu/~judy/domainadapt/
>
> [6] Domain Adaptation: Learning Bounds and Algorithms Mansour et al.

---

> > ### Author Response · Authors · 2021-11-30
> > **Discussion**
> >
> > Dear reviewer,
> >
> > First of all, our apologies for the late reply. There seems to be some problem with the openreview mailing system. Nevertheless, sorry about this as we just noticed your new comment today.
> >
> > Secondly, we would like to continue our discussion regarding your concerns. Please see below:
> >
> > **wrt theoretical results**
> >
> > As mentioned earlier, using the bound directly on the input space might result in a large upper bound of the target loss. However, when **applying the bound on the representation space**, such inflation is neglectable, since the KL term is minimized (both KL and L1 stay near 0 in our experiments). We further prove that under some assumptions, the conditional misalignment in the representation space is bounded by the conditional misalignment in the input space, which is a new contribution and makes the representation alignment method sound. Although in the simple example mentioned, the KL between the input distribution is large, please keep in mind that our KL term is in the representation space, which will be minimized to be very small in practice, making the bound meaningful.
> >
> > Another point is that we recheck some earlier works (e.g., [1]) and the triangle inequality plays an important role in the proof, so we argue that these restrictions were crucial for earlier works and not just for simplicity. We argue that our **much more** general setting (virtually all common loss functions and predictive distributions, not requiring the true labeling function, etc.) can be important for modern domain adaptation.
> >
> > **wrt experimental results and comparison vs modern baselines**
> >
> > We respectfully disagree. We argue that [2] is not a "plain marginal alignment" technique, in the sense that it does not align the marginal distribution of the representation (also the keywords marginal distribution or marginal alignment do not appear in the paper). It minimizes the discrepancy (induced by a classifier/scoring function) between two domains (note that the discrepancy notation $d^{\rho}_{f,\mathcal{F}}(P,Q)$ also depends on the classifier $f$). As you mentioned earlier, using the discrepancy induced by the classifier can make the bound smaller; however, it results in a completely different kind of alignment.
> >
> > Although our baselines are not "modern", we note that almost all advanced DA techniques are still using MMD or DANN whenever they need to align representation distribution. In this paper, we show that reverse KL is a much more efficient and effective alignment technique, which we argue has an important contribution to the field on its own. Note that other reviewers seem to agree with this contribution of our work. Also, note that (quite apparently) [2] cannot be used as the backbone to align representation in those sceneriors.
> >
> > **wrt to additional experimental analysis**
> >
> > Thank you for your recommendation. The representation space of our method is much more aligned compared to other methods (which results in the improved performance). We will certainly add the visualization of the representation space in the camera-ready version (if the paper gets accepted) or in a future submission (if it gets rejected this round).
> >
> >
> > **A final note:** While we agree that our paper has some certain limitations, we urge the reviewer to also consider other contributions of our paper (in both the technical and empirical side) to possibly (and hopefully) improve your rating. We really think that these contributions can be interesting to the community. Thank you for your time reviewing our paper and for your valuable discussions.
> >
> > Best regards,
> >
> > Authors.

---

> > > ### Author Response · Authors · 2021-12-02
> > > **Final thought**
> > >
> > > Dear reviewer,
> > >
> > > We are wondering if you could give some comments and final thoughts about our paper and about our previous discussion.
> > >
> > > Thank you very much.
> > >
> > > Best regards,
> > >
> > > Authors

---

### Official Review · Reviewer_i12Z · 2021-11-03

**Correctness:** 3
**Technical Novelty And Significance:** 3
**Empirical Novelty And Significance:** 3
**Recommendation:** 6
**Confidence:** 3

**Main Review:**

Pros:

1. Marginal distribution alignment is an importance backbone for domain adaptation while the popular adversarial one requires minimax optimization and can be unstable. This paper proposes an reverse KL which can be optimized together with the source classification loss directly.

2. This paper points out a deficiency of Ben David's bound on domain adaption: it fails to consider the problistic labeling mechanism. When the true labeling function is non-deterministic, the bound by David may be inaccurate.


Cons:

1. My major concern is that : the implementation is inconsistent with the derived generalization bound.  I highly appreciate authors' honesty in writing down that they also use the forward KL term as additional penalty, but when $\beta_{aux}$ is large, the objective becomes another one. For instance, in the digits, $\beta=0.3, \beta_{aux}=0.1$, the forward KL term also contributes a lot to the final objective. A very important point is that I didn't find the ablation studies about this term. It would be better if authors could present baseline method (e.g., $\lambda_{aux}=0$) results and show the reverse KL is effective.

2. It can be too strong to assume that $I(z,y)=I(x,y)$ when z has much lower dimension compared to a high dimension input $x$.

Side question:

This paper computes the density by the Gaussian density function and then compute the KL term. I'm wondering how close are the learned representations to the multivariate gaussian distributions, especially for the high-dimensional images, i.e., the Visda datasets.

Update:

I read other reviews and the author reponses, I'm good with authors' reply, especially the KL term experiments. So I raise my score to 6.

**Summary Of The Paper:**

This paper proposes a generalization bound and a reverse KL term to match the marginal distribution.

**Summary Of The Review:**

This paper proposes an reverse KL term to match the marginal distributions in domain adaptation. The results are promising. The main concern is the implementation is inconsistent with the new generalization bound.

---

> ### Author Response · Authors · 2021-11-19
> **Rebuttal**
>
> Dear reviewer,
>
> We appreciate your constructive comments. We agree that marginal distribution alignment is an importance backbone for domain adaptation and can be improved further, which would also benefits other DA algorithms. Please see our answers/clarification below:
>
> **the implementation is inconsistent with the derived generalization bound. It would be better if authors could present baseline method (e.g., $\beta_{aux}=0$)**
>
> We thank the reviewer for your feedback. Indeed, we already have the results for the case of $\beta_{aux}=0$ in an earlier implementation of our model. The results are as follows:
>
> - RotatedMNIST:
>
> | Method | $\mathcal{M}_{15}$ | $\mathcal{M}_{30}$ | $\mathcal{M}_{45}$ | $\mathcal{M}_{60}$ | $\mathcal{M}_{75}$ | Average |
> | --- | --- | --- | --- | --- | --- | --- |
> | KL ($\beta_{aux}=0.0$)| 97.8±0.5| 96.6±0.4| 92.0±0.4| 68.8±4.6 | 62.3±4.2 | 83.5 |
> | KL (reported in paper)| 97.8±0.1| 97.1±0.2 | 93.4±0.8 | 75.5±2.4 | 68.1±1.8 | 86.4 |
>
> - DIGITS:
>
> | Method | M $\rightarrow$ U | U $\rightarrow$ M | S $\rightarrow$ M | Average |
> | --- | --- | --- | --- | --- |
> | KL ($\beta_{aux}=0.0$)| 98.2±0.2| 97.1±0.4 | 90.0±1.1| 95.1 |
> | KL (reported in paper)| 98.2±0.2 | 97.3±0.5 | 92.5±0.9 | 96.0 |
>
> - VisDA17: In this VisDA17 experiment, our method (with $\beta_{aux}=0$) achieves 67.8% accuracy, which is 2.8% lower than the number reported in our paper.
>
>
> Clearly, most of the improvement comes from the reverse KL regularizer term. And even without the auxiliary forward KL term, our results are still significantly higher than the baselines. Note that the auxiliary term requires virtually no extra computation, so we believe it is reasonable to do so. Also, note that the use of auxiliary objectives is not uncommon in the machine learning practice.
>
>
> **It can be too strong to assume that $I(z,y)=I(x,y)$ when $z$ has much lower dimension compared to a high dimension input $x$.**
>
> We respectfully disagree. Although $z$ has a lower dimension compared to $x$, it does not need to contain all the information about $x$, but just the information related to $y$ (hence the mutual information terms $I(z,y)$ and $I(x,y)$). And since $y$ is often very low dimensional, this is likely achievable for the source domain, which the model is trained on. Indeed, this is often what happens in practice, with the empirical evidence being that using $z$ to predict $y$ often yields (near) optimal accuracy in the source domain (indicating that $z$ contains "sufficient" information about $y$).
>
> **This paper computes the density by the Gaussian density function and then compute the KL term. I'm wondering how close are the learned representations to the multivariate gaussian distributions, especially for the high-dimensional images, i.e., the Visda datasets.**
>
> There might be a slight misunderstanding here. We are using the Gaussian distribution for the representation distribution of **each** datapoint $x$, i.e., $p(z|x)$. The marginal distribution of the representation $p(z)$ is the mixture of (infinite number of) Gaussian distributions, and can be of any form. Also note that the Gaussian representation $p(z|x)=\mathcal{N}(z;\mu(x),\text{diag}(\sigma^2(x)))$ is a generalized version of a typical deterministic representation network (it becomes the normal deterministic representation when $\sigma^2(x) \rightarrow 0,  \forall x$). Therefore, this Gaussian representation distribution is not at all restricted when compared to the typical deterministic representation.
>
> Please let us know if you have further follow-up discussions.
>
> Best regards,
> Authors.

---

### Author Response · Authors · 2021-11-23
**General Response: Regarding the revision**

Dear reviewers,

We have updated our paper with a revision. In this version, we add:

- Additional references and related works as suggested by some reviewers.
- Additional analysis on the contribution of the auxiliary forward KL term.
- Experimental results for the Office31 dataset.

Many thanks,

Authors.

---

### Decision · Program_Chairs · 2022-01-20

**Decision:**

Accept (Poster)

**Comment:**

This paper derives a generalization bound on target loss based on training loss and reverse KL divergence between source and target representation distributions. Then, proposes an algorithm for DA using inverse KL on representations. they show that inverse KL term can be estimated efficiently without the need for additional networks and minimax objective. The experiments show the efficiency of the proposed algorithm in terms of improving target accuracy. The paper touches an important problem and the proposed idea is simple and effective.

There were several concerns regarding the paper that were addressed during rebuttal period, such as strength of assumptions, experiments with different values of beta, experiments on office31 dataset, novelty of theoretical results, significance of derived bounds and comparison to [3]. The remaining concern is on comparing the proposed method to recent work in domain adaptation.

I ask the authors to add the following to the camera ready (1) visualizations they have promised that depicts their method leading to better alignment and (2) add the points raised in defending the novelty of theoretical results.